EMBO
*reports*

# Neuralized-like proteins differentially activate Notch ligands

Alina Airich[1,3], Oren Gozlan [2,3], Ekaterina Seib[1], Gittel Leah Shaingarten[2], Lena-Sophie Wilschrey[1], Liora Lindenboim[2], David Sprinzak [2✉] & Thomas Klein [1✉]

## Abstract

**Notch signalling is a major signalling pathway coordinating cellular processes between neighbouring animal cells. In *Drosophila*, two E3 ubiquitin ligases, Neuralized (Neur) and Mindbomb1 (Mib1), regulate Notch ligand activation and are essential for development. However, the mammalian orthologs of Neur, Neuralized-like (NEURL) 1 and 1B, appear to be dispensable for development, as double knock-out mice show no overt developmental defects. Thus, it is unclear if and how NEURL proteins regulate the mammalian Notch ligands. To address this question, we examined NEURL proteins' ability to activate Notch ligands in a humanized *Drosophila* model and mammalian cell culture. We found that, unlike MIB1, NEURL proteins activate Notch only with a subset of mammalian ligands, which contain a Neuralized binding motif. This motif has the consensus sequence NxxN and is present only in Notch ligands DLL1 and JAG1, but not in DLL4 and JAG2. Thus, our results reveal a differential regulatory mechanism of Notch activation in mammals, which can potentially explain the limited role of NEURL proteins in mammalian development and homeostasis.**

**Keywords** Notch signalling; Neuralized; E3 ubiquitin ligases; DSL ligands
**Subject Category** Signal Transduction

## Introduction

Notch signalling is a highly conserved pathway that coordinates cellular processes between neighbouring cells throughout the animal kingdom (Kovall et al, 2017; Siebel and Lendahl, 2017; Sprinzak and Blacklow, 2021). Notch signalling relies on the interaction between membrane-bound Delta/Serrate/Lag2 (DSL) ligands on a sender cell and Notch receptors on a receiver cell. While highly conserved, there are differences between the number and type of Notch ligands and receptors across animal species. In *Drosophila*, there are two ligands, termed Delta (Dl) and Serrate (Ser), and one Notch receptor. In mammals, there are five canonical

ligands, three from the Delta-like (Dll) family (Dll1, Dll3, Dll4), and two from the Jagged (Jag) family (Jag1 and Jag2). There are four mammalian Notch receptors (Notch1-4). All ligands and receptors can interact with each other, albeit with different signalling strengths (Kovall et al, 2017; Siebel and Lendahl, 2017). Yet, it is unclear how different receptor-ligand interactions are specifically regulated to induce different cellular outcomes.

Canonical Notch signalling is initiated by the binding of DSL ligands in the sender cell to the Notch receptor in the receiver cell (Fig. 1A). Following ligand binding, a pulling force exerted on the extracellular domain of Notch induces a conformational change in the receptor. This conformational change facilitates sequential proteolytic cleavages, first by ADAM10 (S2-cleavage) and then by the γ-secretase complex (S3-cleavage), releasing the Notch intracellular domain (NICD) into the cytosol. NICD subsequently translocates from the cytosol to the nucleus, where it regulates the expression of target genes.

The pulling force on the Notch receptor is generated by ligand endocytosis, which is triggered by the ubiquitylation of the ligand's intracellular domain mediated by specific E3 ligases. In *Drosophila*, two E3 ligases, termed Neuralized (Neur) and Mindbomb1 (Mib1), are involved in the activation of Notch ligands in the sender cells (Rullinkov et al, 2009; Weinmaster and Fischer, 2011; Seib and Klein, 2021), (Fig. 1B). Both ligases share no overall sequence or structural similarity beyond the RING domains at their C-terminus (Fig. 1B). Nevertheless, both induce endocytosis of the ligands either by ubiquitylation (ubi) of lysine residues found on the ligand's intracellular domain (ICD) or, in the case of Neur, also in a ubi-independent manner (Berndt et al, 2017; Troost et al, 2023b). For activation, the two E3 ligases bind the ligands at different epitopes within the ICD. MIB1 was shown to bind in a bipartite mode at different epitopes of the ligands' ICD (Daskalaki et al, 2011; McMillan et al, 2015). *Drosophila* Neur binds its substrates at the Neuralized-binding motif (NBM) with the consensus sequence NxxN (Bardin and Schweisguth, 2006; Fontana and Posakony, 2009). This NxxN sequence is conserved across all known Drosophila Neur substrates, including Ser, Dl, and the Bearded proteins. Whether human NEURL similarly utilizes the NxxN motif for effective Notch activation remains an open question.

In mammalian genomes, one Mib1 ortholog and two Neur orthologs, Neuralized-like-1 (Neurl1), and Neuralized-like-1B

[1]Institut fuer Genetik, Heinrich-Heine-Universitaet Duesseldorf, Universitaetstr. 1, 40225 Duesseldorf, Germany. [2]School of Neurobiology, Biochemistry, and Biophysics, The George S. Wise Faculty of Life Sciences, Tel Aviv University, Tel Aviv 69978, Israel. [3]These authors contributed equally: Alina Airich, Oren Gozlan. ✉E-mail: davidsp@tauex.tau.ac.il; thomas.klein@hhu.de

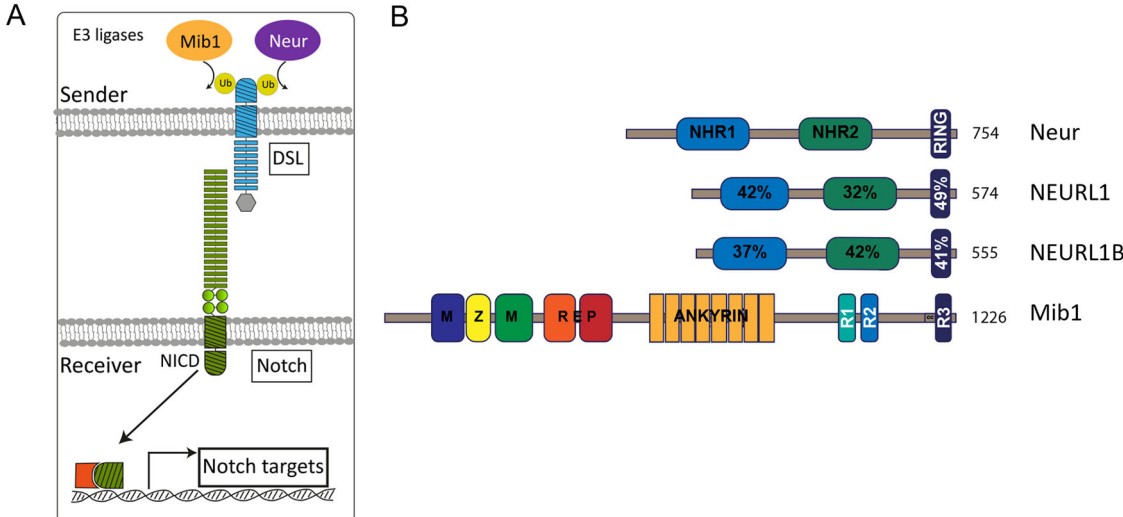

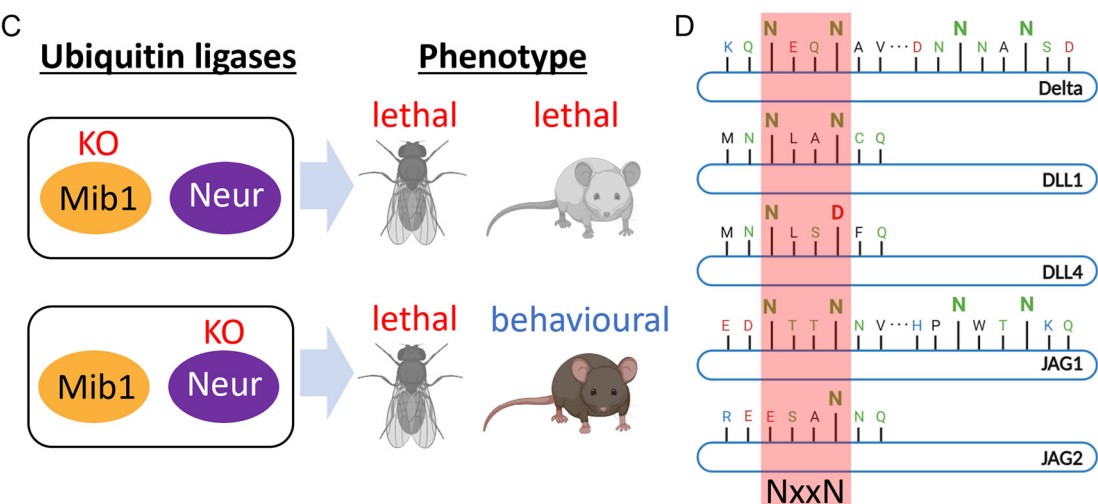

**Figure 1. The role of Neur and NEURL proteins in ligand-dependent activation of the Notch pathway.**

(A) E3-ligases Mib1 and Neur mediate ubiquitylation of the ICDs of DSL ligands in the sender cell, which in turn leads to activation of Notch receptors in the receiver cell. (B) The structure of Neur, NEURL1, NEURL1B, and MIB1. The Neur family members have two NHR-domains, followed by a RING domain. MIB1 consists of the MZM and REP domains, followed by ANK repeats and three RING domains. (C) The distinct KO-phenotypes of MIB1 and Neur in *Drosophila* and mouse. (D) The position of the NxxN consensus sequence in the ICDs of the mammalian ligands. The NxxN motif is found in Dl, DLL1, and JAG1. DLL4 and JAG2 appear to have a cryptic sequence where one N is replaced by D or E, respectively. The sequences are conserved among mammalian orthologs (see Fig. EV1).

(Neurl1B, also known as Neur2), have been identified. A knock-out (KO) of Mib1 in mice is lethal and causes the expected Notch-like developmental phenotypes (Koo et al, 2005) (Fig. 1C). In contrast, Neurl1 and -1B double KO mice survive to adulthood without any obvious developmental defects (Song et al, 2006) (Fig. 1C). This has led to the general belief that only Mib1, but not Neurl1 and -1B, regulates Notch ligand activity in mammals. Nevertheless, more recent work indicates that Neurl proteins can bind to the ICD of ligands and in some cases suppress Notch signalling in cell culture assays (Koutelou et al, 2008; Rullinkov et al, 2009).

Additional analysis in adult mice of the Neurl1 and -1B of single and double KOs revealed that their function is largely restricted to postnatal cognitive processes, such as synaptic plasticity and spatial memory, consistent with their expression patterns particularly in regions like the cerebral cortex and hippocampus (Ruan et al, 2001; Timmusk et al, 2002; Pavlopoulos et al, 2011; Lee et al, 2020). Interestingly, conditional KO of Jag1 and Notch1 in the brain has also been associated with spatial memory impairments (Wang et al, 2004; Sargin et al, 2013; Lee et al, 2020). Given the similarities, Neurl proteins in mammals appear to have a specialized role in Notch signalling, which is largely confined to specific neuronal processes in adults, rather than general development. In contrast, *Drosophila* Neur is essential both for development and neural-specific processes, including memory formation and other behaviours (Morel and Schweisguth, 2000; Lai et al, 2001; Pavlopoulos et al, 2008; Lieber et al, 2011). Thus, an intriguing evolutionary divergence exists, where mammalian Neurl proteins appear to have lost their developmental function but retained a specialized role in neuronal regulation.

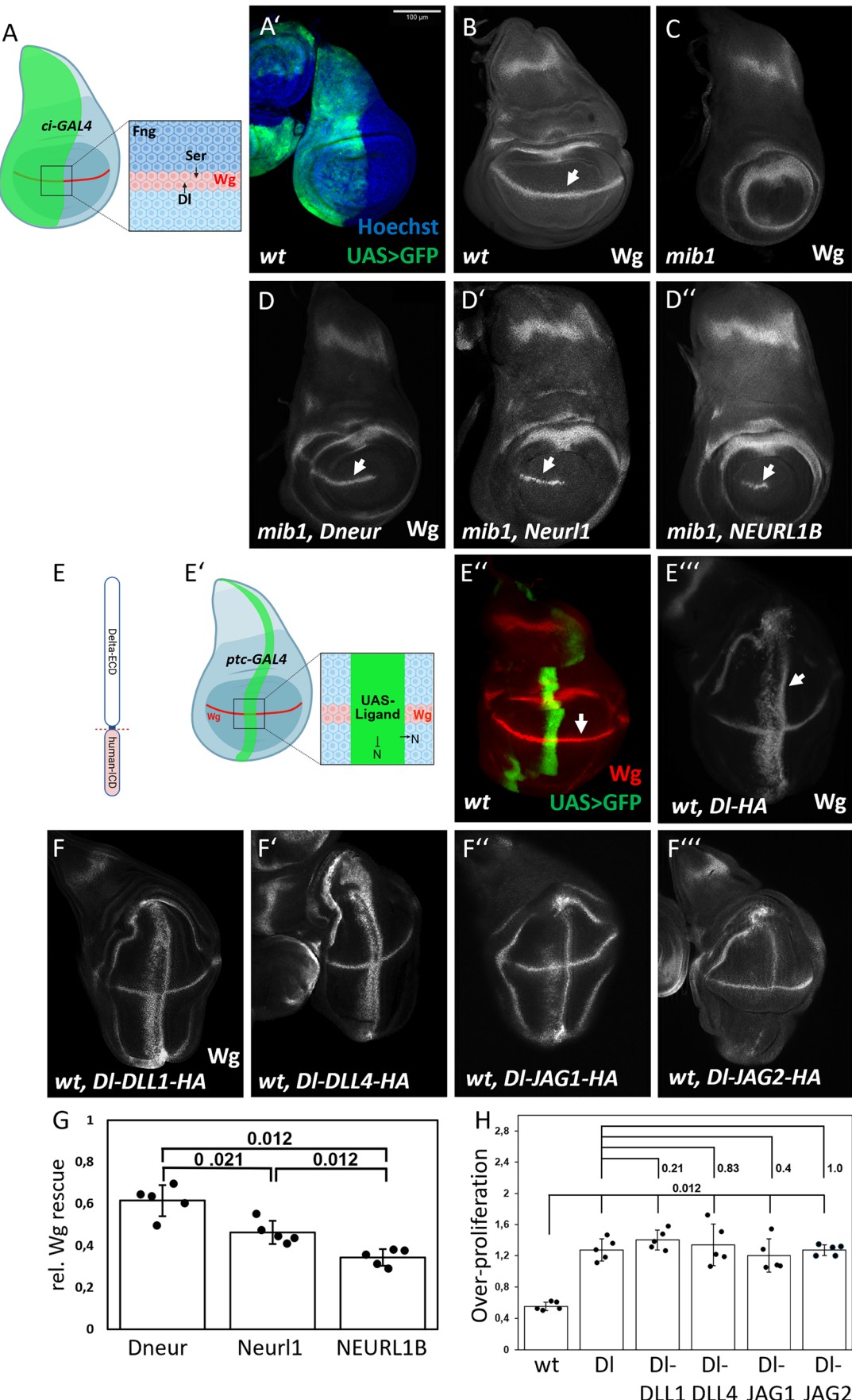

**Figure 2.   Activation of endogenous and human Notch ligands by Neur, Neurl1 and NEURL1B in the wing imaginal disc in *Drosophila*.**

(A) Schematic of the expression domain of ci-GAL4, which is expressed throughout the anterior compartment (green area). At the D/V boundary, interaction between dorsal and ventral boundary cells, mediated by Dl and Ser activate the Notch pathway and induce the expression of target genes, such as Wg. (A') ci-Expression domain visualized by the expression of UAS-GFP of the third instar wing imaginal disc. (B) Wild-type Wg expression with Notch-dependent expression at the D/V boundary (arrowhead). (C) In mib1 mutant disks, the expression of Wg along the D/V boundary is lost. The wing pouch is significantly reduced. (D-D") Expression of Neur, Neurl1, and NEURL1B in mib1 mutant wing disks results in the re-establishment of Wg expression along the D/V boundary (arrowhead in D-D"). (E) Schematic of humanized hybrid ligands consisting of the ECD and TM of Dl and the ICD of each canonical ligand. (E') Schematic of ptc-Gal4 expression in the wing imaginal disc. (E") ptc-Gal4 is expressed in a broad stripe at the anterior side of the A/P compartment boundary. The expression domain runs perpendicular to the expression of Wg along the D/V boundary (arrowhead). (E''') Expression of Dl with ptc-GAL4 results in the induction of two ectopic stripes of Wg expression (arrowhead). (F-F''') Expression of Dl-DLL1, Dl-DLL4, Dl-JAG1, and Dl-JAG2 with ptc-Gal4 results in the ectopic expression of Wg and over-proliferation. (G) Quantification of Wg-rescue (D-D") upon Neur and NEURL expression in mib1-mutant disks. Wg-rescue was determined by measuring the length of Wg-expressing cells at the D/V boundary (see Methods). (H) Quantification of over-proliferation (F-F'''). Over-proliferation was determined by dividing the diameter along the anterior-posterior direction with the diameter along the dorso-ventral direction of the wing pouch (A/P:D/V, see Methods). Error bars represent the standard error. Denoted *p* values were calculated by the Mann–Whitney *U*-test (*n* = 5). Source data are available online for this figure.

Here, we investigated the ability of human/mouse NEURL1 and NEURL1B to activate the human Notch ligands in both humanized *Drosophila* and mammalian cell culture. We found that both NEURL proteins can activate a subset of the ligands. The ability of NEURL proteins to activate ligands depends on the presence of the NBM sequence (NxxN) in their ICD. Ligands that contain this NBM in their ICD, DLL1, and JAG1, can activate Notch receptors in a NEURL-dependent manner both in vivo (*Drosophila*) and in vitro (cell culture assay). Ligands lacking the NBM, DLL4, and JAG2, cannot activate NEURL-mediated Notch signalling in either system. Moreover, mutating the NBM in the ICD of DLL1 and JAG1 abolishes NEURL-mediated Notch activity but not MIB1-mediated activity. Introducing an NBM in the ICD of DLL4 rescued NEURL activity in both *Drosophila* and mammalian cell culture. Moreover, we found increased co-localisation of NEURL1 and Notch ligands only in ligands activated by NEURL1 (DLL1 and JAG1), suggesting stronger interaction of NEURL1 with these ligands. Thus, our results indicate that NEURL1 and NEURL1B can differentially activate ligands containing the NBM (DLL1 and JAG1), while showing no activation of the other ligands (DLL4 and JAG2). We suggest that the cognitive phenotypes associated with Neurl1 and -1B in mice may be due to Notch-dependent processes restricted to specific ligands.

## Results

A sequence comparison of Dl with the mammalian DSL-ligands revealed that the ICDs of Dll1 and Jag1 in several vertebrate species contain a sequence at the N-terminus that matches the NxxN consensus motif of an NBM at a position comparable to the characterized NBM of the ICDs of the *Drosophila* ligands (NLAN in DLL1 and NTTN in JAG1, Figs. 1D and EV1). The strong conservation suggests that the NxxN motif undergoes evolutionary selection, indicating its potential importance for the function of the ligands. While the DLL4-ICD contains a sequence similar to DLL1's NBM, it does not have a complete NxxN motif, as the second N is replaced by a negatively charged aspartic acid (D) (NLS**D**, Figs. 1D and EV1). Likewise, in the case of JAG2, a sequence showing similarity to the NxxN motif exists, where one N is replaced by a negatively charged glutamic acid (E) (**E**SAN) (Figs. 1D and EV1).

## Neurl1 and NEURL1B can activate the Notch pathway in *Drosophila*

To determine whether NEURL proteins function as orthologs of Neur and can activate specific ligands, we analysed their activity in complementary experiments using *Drosophila* and mammalian cell culture. Due to the limited availability of human NEURL1, we used mouse Neurl1 and human NEURL1B for our *Drosophila* experiments. Mouse Neurl1 shares 94.3% sequence identity with human NEURL1 (Rice et al, 2000), making it a practical alternative to NEURL1. *Drosophila* experiments showed NEURL activity in an in vivo setting, while mammalian cell culture assays allowed quantitative assessment in an in vitro setting.

We first asked whether Neurl1 and NEURL1B can activate endogenous *Drosophila* ligands. To address this, we expressed Neurl1-myc and NEURL1B-myc in *mib1* mutant wing imaginal disks using *ci*-Gal4, which drives expression of UAS constructs throughout the anterior compartment (Fig. 2A-A'). As a readout for Notch activity, we used the expression of *wingless* (Wg) along the dorso-ventral compartment boundary (D/V boundary) (Fig. 2B, arrowhead). The expression of Wg depends on the activity of both ligands, with Ser being the dominant one (Klein, 2001; Troost and Klein, 2012). In this genetic context, the expressed Neur- and NEURL-variants are the only E3 ligases present in the disks that can activate the Notch pathway. The constructs used here are inserted into the same genomic landing site to ensure comparable expression levels (Appendix Fig. S1A-A"). The loss of *mib1* function results in a strong loss of Notch activity revealed by the loss of Wg along the D/V boundary and a dramatic size reduction of the wing primordium (Lai et al, 2005; Le Borgne et al, 2005) (Fig. 2C, compared to Fig. 2B). We performed our experiments in a *mib1* single mutant background rather than a *mib1 neur* double mutant, because *neur* expression occurs only late in wing development within a limited subset of cells (sensory organ precursors), and its complex promoter architecture restricts experimental manipulation. Moreover, the ubiquitous expression of Mib1 makes it impossible to completely discern the function of the two E3 ligases, especially in Neur-dependent processes. We found that the expression of Wg along the D/V boundary is re-established in the region of expression of both *Drosophila* Neur and both NEURL proteins (Fig. 2D-D", arrow, compare with B and C). There are differences in the degree of rescue of Notch activity

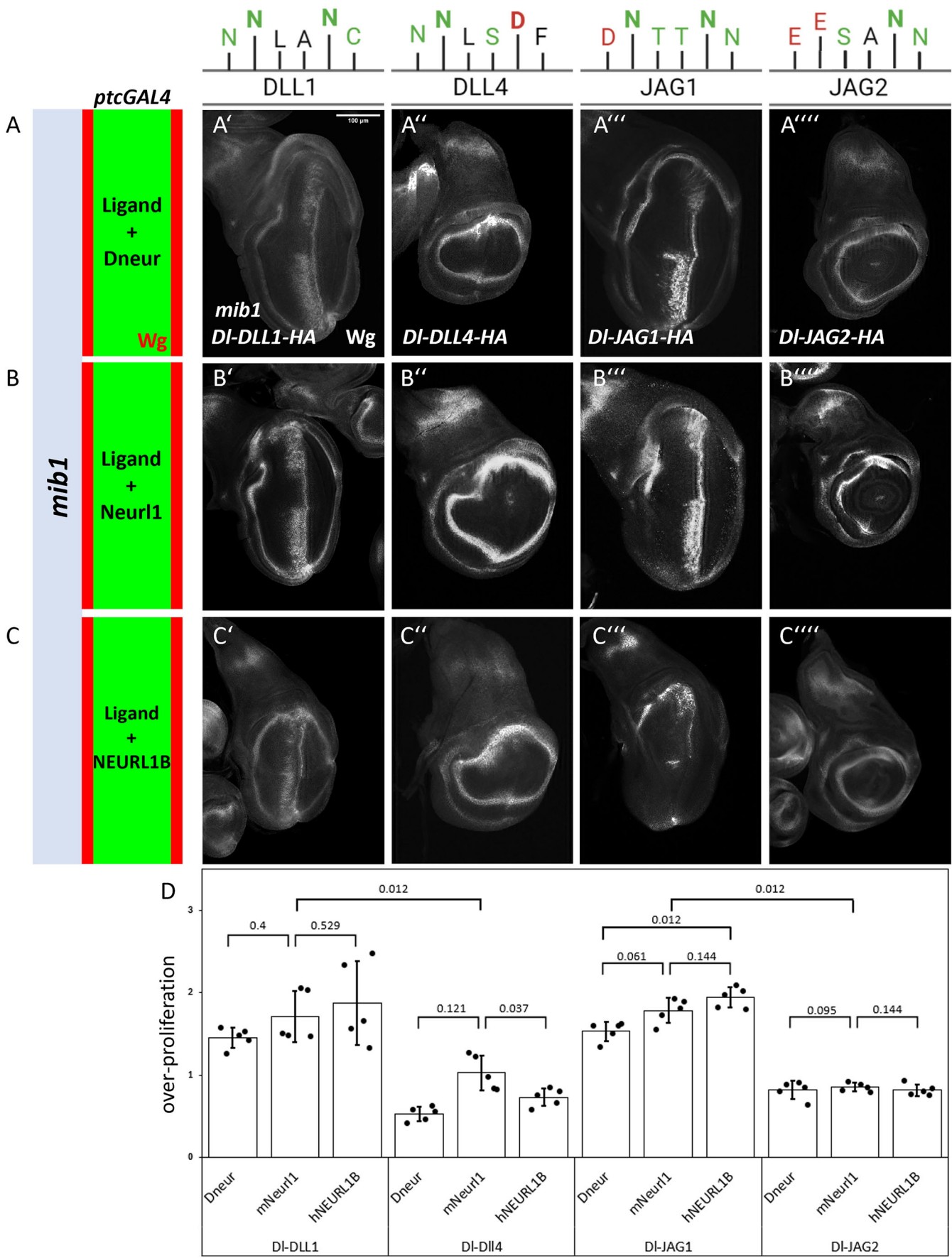

Figure 3. Activation of the mammalian ICDs of DSL ligands by Neur, Neurl1 and NEURL1B.

(A–C"") Co-expression of hybrid ligands and Neur variants in mib1 mutant disks. Only Dl-DLL1 and Dl-JAG1, which contain the NxxN sequence, can be activated by Neur, Neurl1, and NEURL1B. (D) Quantification of over-proliferation induced by co-expression of hybrid ligands and Neur proteins in mib1-mutant disks. Over-proliferation measure was determined by dividing the diameter along the anterior-posterior direction with the diameter along the dorso-ventral direction of the wing pouch (A/P:D/V, see methods). Error bars represent the standard error. Denoted *p* values were calculated by the Mann–Whitney *U*-test (*n* = 5). Source data are available online for this figure.

between the two NEURL proteins compared to Neur. Both NEURL proteins exhibit weaker activation compared to Neur, with Neurl1 showing better rescue than NEURL1B (length of DV stripe in Fig. 2D-D", quantification in Fig. 2G). Hence, the results indicate that each of the two NEURL proteins can activate endogenous *Drosophila* ligands to a level sufficient to induce the expression of Wg, which requires high Notch activity.

## The ICDs of the human ligands can mediate Mib1-dependent activation of Dl-hybrids in *Drosophila*

To rigorously test the activity of the ICDs of mammalian ligands in *Drosophila*, we generated hybrid ligands in which the ICD of Dl is replaced by that of human ligands (Dl-DLL1, Dl-DLL4, Dl-JAG1, and Dl-JAG2, Fig. 2E). We excluded the ICD of DLL3 since it is not a ligand capable of trans-activation of Notch (Ladi et al, 2005; Geffers et al, 2007). The constructs were inserted into the same genomic landing site and expressed with the GAL4 system.

We first expressed the Dl-hybrids in wild-type disks using *ptc*-GAL4, which drives expression visible as a stripe along the anterior side of the anterior-posterior compartment boundary (A/P-boundary, Fig. 2E',E"). Equivalent expression of the ligands was confirmed by HA staining (Appendix Fig. S1B-B").

Expression of Dl with *ptc*-GAL4 induces the expression of Wg in two ectopic stripes running perpendicular to the endogenous expression domain that straddles the D/V boundary (Fig. 2E"'). Ectopic expression of all four Dl-hybrids with *ptc*-GAL4 resulted in similar ectopic activation of the Notch pathway in the wing disc, as indicated by the induction of over-proliferation and Wg expression (Fig. 2F-F"', quantification in Fig. 2H). While the expression of Dl-DLL1, Dl-DLL4, and Dl-JAG1 resulted in the Notch-pathway activation at a level comparable to Dl, there was less ectopic expression of Wg in the case of Dl-JAG2, indicating that the ICD of JAG2 is a less potent activator of Notch (Fig. 2F-F"', compare with E"', quantification in Appendix Fig. S1C). None of the hybrid ligands were able to activate the Notch pathway in *mib1* mutant disks (Appendix Fig. S1D-D"'). Given that Mib1 is a central E3 ligase ubiquitously expressed in the wing disc at this stage, these results indicate that the hybrid ligands are capable of inducing Mib1-mediated Notch activation in *Drosophila* (Lai et al, 2005; Le Borgne et al, 2005; Wang and Struhl, 2005).

## Neurl1 and NEURL1B differentially activate hybrid DSL-ligands in humanized *Drosophila*

Having confirmed that the hybrid ligands are functional in *Drosophila*, we next wanted to test the ability of Neur and its mammalian orthologs to activate the hybrid constructs. As such, we co-expressed the hybrid constructs with the NEURL proteins in *mib1* mutant disks. In this genetic setting, the NEURL proteins are the only E3 ligases present to activate the Notch ligands. Co-expression of Dl with Neur in *mib1* mutants results in the induction of two ectopic stripes of Wg (Berndt et al, 2017). We found that activation of Notch was observed also upon co-expression of either Neur, Neurl1 or NEURL1B with Dl-DLL1 or Dl-JAG1, but not with Dl-DLL4 or Dl-JAG2 (Fig. 3A–C"", quantification in Fig. 3D). This matched our sequence comparison that showed that DLL1 and JAG1, but not DLL4 and JAG2, contain the NxxN motif. The Wg activation of Dl-JAG1 and Dl-DLL1 induced by Neurl1 was stronger than that induced by NEURL1B (Fig. 3B, C"", quantification in Appendix Fig. S2A). This suggests that Neurl1 is a better activator of DSL-ligands than NEURL1B. We found that also a Dl-hybrid containing the ICD of rat Dll1 (Dl-rDll1), which contains the conserved NxxN sequence, could be activated by all three Neur proteins and also Mib1 in a manner indistinguishable from human Dl-DLL1 (Appendix Fig. S2B-B""). This validates the broad activation of Dll1 family members by the NEURL proteins. Overall, our findings demonstrate that Neur and its mammalian orthologs selectively activate humanized hybrid ligands.

## NEURL1 differentially activates DSL-ligands in mammalian cells

We next wanted to test whether NEURL1 differentially activates DSL-ligands in mammalian cell culture. To assess NEURL1 activity, we performed a Notch activity assay by co-culturing sender cells expressing full-length human DSL ligands (DLL1, DLL4, JAG1, and JAG2) and human NEURL1 with receiver cells expressing a Notch reporter (Fig. 4) (Khamaisi et al, 2022). We used human bone osteosarcoma cells (U2OS) from which MIB1 was knocked out as sender cells (MIB1-KO cells) (Cao et al, 2024). As receiver cells, we used U2OS cells stably expressing hybrid NOTCH-Gal4 receptors, and transfected them with a UAS-Luciferase reporter gene (Gordon et al, 2009; Khamaisi et al, 2022). Notch activity in the receiver cells leads to the release of the intracellular Gal4, which activates the reporter gene (Fig. 4A). We used NOTCH1-Gal4 receiver cells to test DLL1 and DLL4 activity and NOTCH2-Gal4 receiver cells to test JAG1 and JAG2 activity (Gordon et al, 2009), since JAG1 is a very weak activator of Notch1 receptors (Kuintzle et al, 2025). All ligands were able to activate Notch when expressed in wild-type (WT) U2OS cells that contain endogenous MIB1 (Fig. 4B–E, white bars). MIB1-KO abolished the activity of all ligands (Fig. 4B–E, grey bars), showing that MIB1 is the predominant E3 ligase in U2OS cells and that it is required for Notch ligand activity. When NEURL1 was co-expressed with Notch ligands in MIB1-KO cells, we observed induction of Notch activity only with sender cells expressing DLL1 and JAG1, but not with DLL4 and JAG2 (Fig. 4B–E, grey bars). These results indicate that only ligands containing the NxxN motif could be activated by NEURL1, in agreement with the *Drosophila* experiments and the prediction from our sequence analysis.

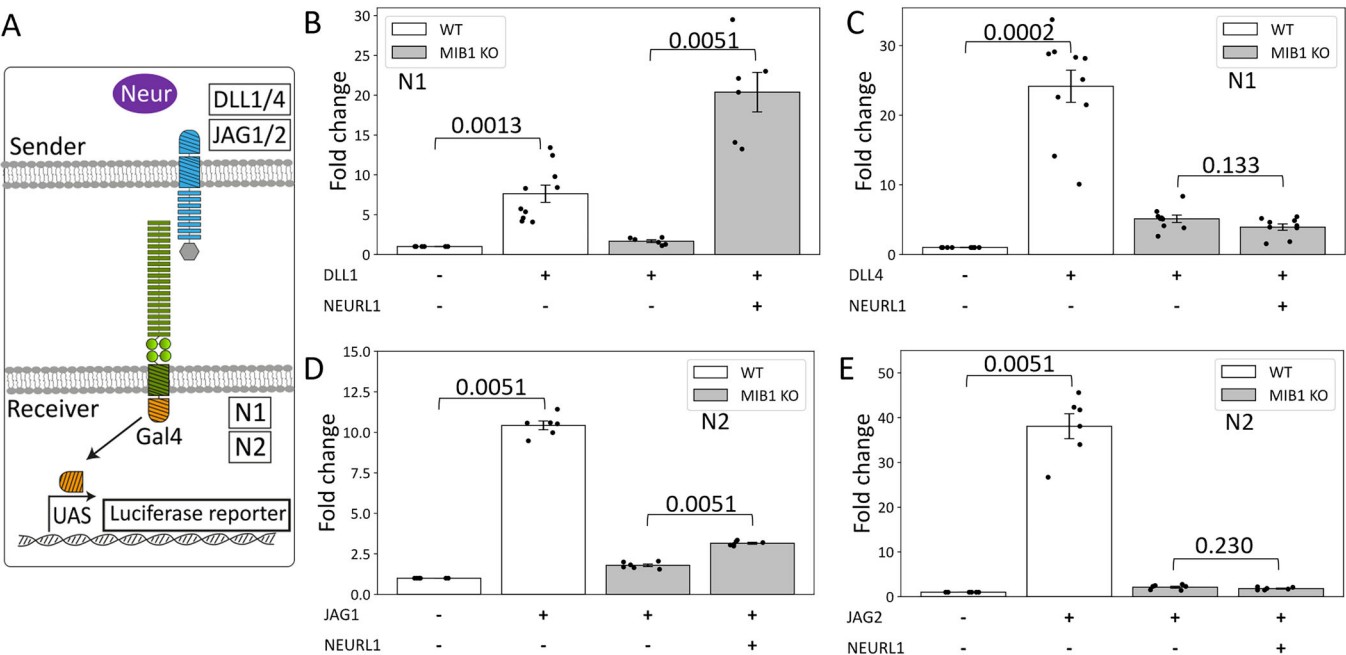

**Figure 4. NEURL1 activates DLL1 and JAG1, but not DLL4 and JAG2, in Mammalian cell culture assay.**

(A) Schematic of the luciferase activity assay. DLL1/4 were co-cultured with NOTCH1-Gal4, while JAG1/2 were co-cultured with NOTCH2-Gal4. 24 h after being co-cultured, the cells were lysed and their Notch activity was measured. (B–E) Luciferase activity assay results for the different Notch ligands. Fold change in luciferase induction in WT cells (white bars) compared to MIB1 KO cells (grey bars) with or without NEURL1 expression. For each plot, the results represent the fold change compared to the negative control. Error bars represent the standard error. Denoted $p$ values were calculated by the Mann–Whitney test ($n \geq 3$). Source data are available online for this figure.

We also attempted to test the ability of NEURL1B to activate Notch ligands in cell culture. However, we found that over-expression of NEURL1B was toxic to the cells, limiting their viability over the time period required for the assay. Nevertheless, NEURL1B was able to activate DLL1, rDll1 and JAG1, but not DLL4 and JAG2 in the *Drosophila* assay.

## The NxxN motif is required and sufficient for activation of Notch ligands by NEURL proteins

After establishing the differential activity of NEURL proteins, we wanted to test whether the NxxN motif was required for NEURL-mediated Notch activity in *Drosophila*. To do so, we exchanged one of the N's in the NxxN consensus sequences of the ICDs of DLL1 and JAG1 to D or A (Fig. 5A). For Dl-DLL1, we chose to exchange the second N in the NBM motif to D to match the cryptic sequence in the DLL4-ICD (as discussed below). For Dl-JAG1, we chose to exchange the first N to A, to avoid disrupting an NN motif on the C-terminus of the NxxN motif, which has been shown to be weakly involved in deltaD signalling in *Zebrafish* (Palardy and Chitnis, 2015). Both variants activate Notch in wing disks that express WT Mib1 (Fig. 5B'-B"). However, in contrast to Dl-DLL1 and Dl-JAG1, the activation of Dl-DLL1-N2D and Dl-JAG1-N2A was diminished upon co-expression with Neurl1 in *mib1* mutant disks (Fig. 5C'-C", quantification in Fig. 5D). Moreover, Dl-rDll1 variants with the complete NxxN motif mutated to A or only the C-terminal N to A or D, could not be activated by Neurl1 (D-rDll1-NBM2A, Dl-rDll1N2A, and Dl-rDll1-N2D, Appendix Fig. S3A-A"). Similarly,

Neurl1 failed to activate Dl-JAG1 variants where either the NxxN motif is completely mutated, or only the N-terminal N was mutated (Dl-JAG1-NBM2A and Dl-JAG1-N2A, Appendix Fig. S3B). These results indicate that the NxxN motif is essential for the activation of DLL1 and JAG1 orthologs by NEURL-family members and that it is a functional NBM.

To further test the requirement for the NxxN sequence in DLL1 and JAG1 as NBM, we analysed the activity of NxxN variants in a cell culture assay. In this assay, we used sender cells expressing full-length ligands containing the same substitutions as above, namely, DLL1-N2D and JAG1-N2A and compared them to WT ligands. Luciferase activity assay showed that these variants can activate Notch in WT cells (containing MIB1) but cannot activate Notch in MIB1-KO cells expressing NEURL1 (Fig. 5E,F). These results show that full-length DLL1 and JAG1 require the NxxN motif to be activated by NEURL1. Thus, it is likely that these motifs are functional NBMs, as found for the binding partners of Neur in *Drosophila*.

## Re-activating the cryptic NBM in DLL4

Sequence comparison revealed a cryptic NBM in DLL4, similar to that of DLL1, but with the C-terminal asparagine (N) substituted by an aspartic acid (D), resulting in the sequence NLS<u>D</u> instead of NLA<u>N</u>. This N-to-D substitution is conserved across all mammals and in chicken (Fig. EV1), while zebrafish show a similar substitution with glutamic acid (E). To test if this substitution from N to D is responsible for DLL4's inability to be activated by

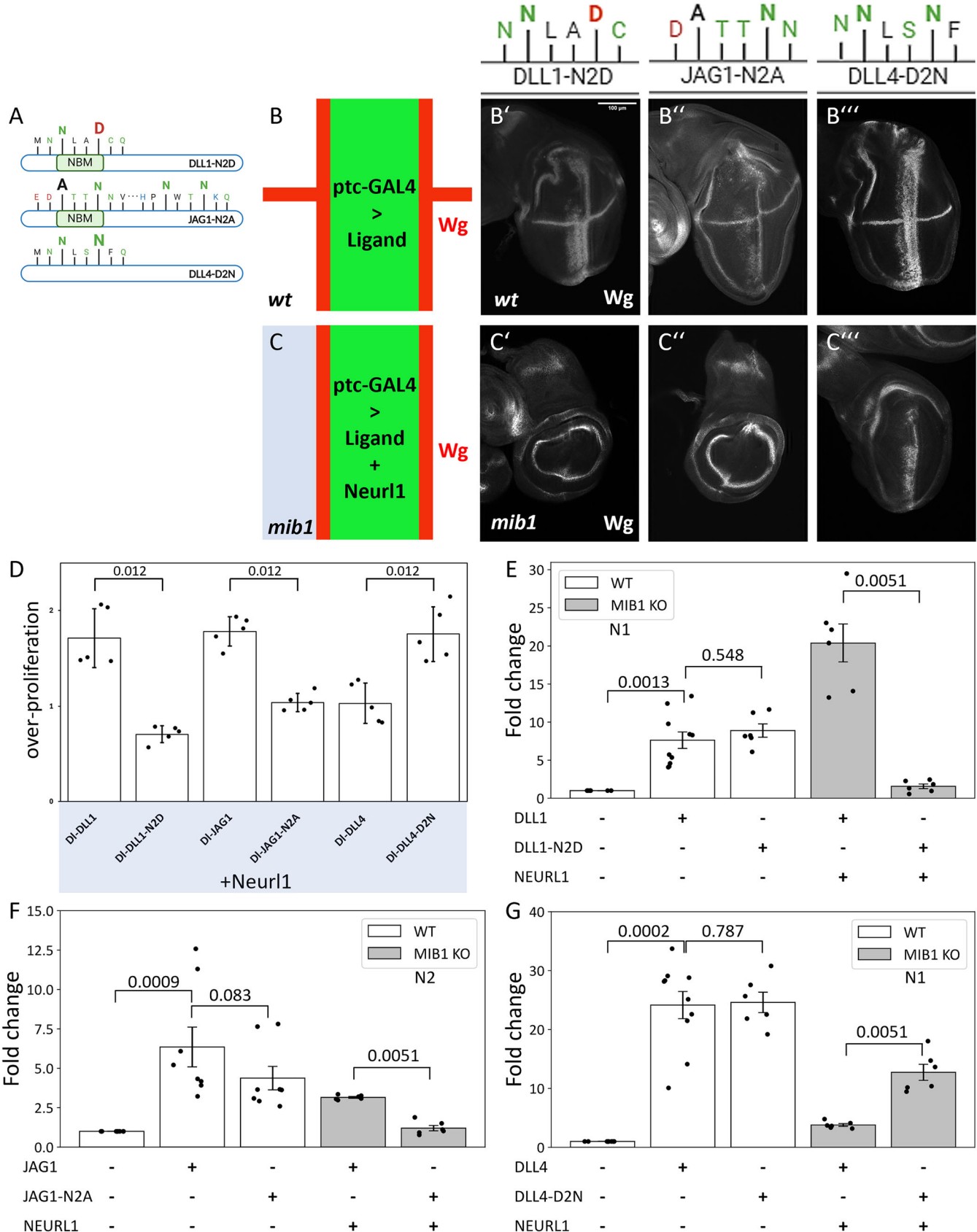

Figure 5. The NxxN motif is required and sufficient for activation of Notch ligands by NEURLs.

(A) Schematic of the variant ICDs used in the experiment, with specific substitutions abolishing the NxxN motif in DLL1 and JAG1 ICDs, while reconstituting it in DLL4. (B-B''') Expression of the mutated Dl-DLL1-N2D, Dl-JAG1-N2A, and Dl-DLL4-D2N in WT wing imaginal disks induces ectopic Wg expression, similar to unmutated Dl-DLL1, Dl-JAG1, and Dl-DLL4 (Fig. 2F-F''). (C-C''') Co-expression of Dl-DLL1-N2D and Dl-JAG1-N2A with Neurl1 in mib1 mutant disks failed to activate Wg, while co-expression of Dl-DLL4-D2N with Neurl1 rescues Wg expression (compare with Fig. 3B''). (D) Quantification of over-proliferation of disks shown in (C-C''') compared to wt hybrid ligands shown in Fig. 3. Over-proliferation was determined by dividing the diameter along the anterior-posterior direction with the diameter along the dorso-ventral direction of the wing pouch (A/P:D/V, see methods). (E-G) Luciferase activity assay results for mutated Notch ligands. Fold change in luciferase induction is measured for cells containing DLL1-N2D (E), JAG1-N2A (F), or DLL4-D2N (G) with or without NEURL1 expression. For each plot, the results represent the fold change compared to the negative control. In WT cells (white bars), all mutated ligands show similar activation to WT ligands. In MIB1-KO cells (grey bars), DLL1-N2D and JAG-N2A show reduced activation by NEURL1, while DLL4-D2N shows enhanced activation by NEURL1. Results represent fold change compared to the negative control. Error bars represent the standard error. Denoted p values were calculated by the Mann–Whitney test (n = 5 for D, n ≥ 3 for E-G). Source data are available online for this figure.

NEURL proteins, we mutated the C-terminal D of DLL4 to N (NLS<u>N</u>) (Fig. 5A), creating the variant DLL4-D2N.

Expression of Dl-DLL4-D2N in wt *Drosophila* wing disks induced ectopic expression of Wg comparable to Dl-DLL4, indicating that the D-to-N mutation does not affect Mib1-mediated signalling (Fig. 5B'''). However, unlike Dl-DLL4, the Dl-DLL4-D2N variant successfully activated Notch signalling in *mib1* mutant disks upon co-expression with Neurl1. This demonstrates that the presence of the second asparagine residue in the cryptic NBM is crucial for DLL4 activation by Neurl1 (Fig. 5C''').

To examine if the NxxN motif alone is sufficient for ligand activation in mammalian cells, we performed cell culture assays comparing full-length DLL4-D2N and DLL4 (Fig. 5G). When expressed in WT U2OS cells (which contain MIB1), both DLL4 variants activated Notch similarly. However, when co-expressed with NEURL1 in MIB1-KO cells, DLL4-D2N showed significantly higher activation than wild-type DLL4, suggesting that restoring the canonical NxxN motif is sufficient to enable NEURL-mediated Notch activation in both *Drosophila* and mammalian contexts.

## Reduced co-localisation between NEURL1 and Notch ligands is observed when the NxxN motif is mutated

Following the activity assays, which demonstrated a dependency on the NxxN motif, we examined whether NEURL1 and the ligands localise to shared subcellular regions. To address this, we conducted co-localisation assays using fluorescent tags fused to the C-terminus of the ligands (mTurquoise2 and mTQ2) and NEURL1 (mCherry, Fig. 6A). As a measure of co-localisation, we defined a co-localisation ratio as the fraction of mCherry (fused to NEURL1) pixels that co-localised with mTQ2 (fused to ligands) pixels. This spatial readout serves as a complementary approach to the activity assays, helping determine whether the observed functional interaction is supported by physical proximity within the cell (Fig. 6B–D).

Image analysis revealed that NEURL1, when co-expressed with DLL1 or JAG1, localises primarily in vesicles that strongly overlap with the two ligands. In contrast, NEURL1 was more evenly distributed in the cytoplasm with significantly reduced co-localisation when co-expressed with Dll4. Remarkably, mutation of the NBM in DLL1 and JAG1 showed a significant decrease in co-localisation, implying that the NBM is required for co-localization (Fig. 6B–C'). Furthermore, looking at the DLL4 variant in which the NBM has been re-introduced (DLL4-D2N) we found a significant increase in its co-localisation with NEURL1 (Fig. 6D, D'). Together, these results suggest that the NBM contributes to the

interaction of NEURL1 with the Notch ligands, regulating its subcellular localisation.

## The NBM is not required for co-immunoprecipitation of NEURL1 and Notch ligands

To test whether the NBM motif mediates the binding between NEURL1 and the different Notch ligands, we performed co-immunoprecipitation (co-IP) assays. NEURL1-mCherry was tagged with a C-terminal FLAG epitope, and each ligand variant was tagged with an HA epitope. FLAG-tagged NEURL1 served as bait for immunoprecipitation, and co-precipitated ligands were detected by anti-HA Western blot analysis (Appendix Fig. S4). Surprisingly, all tested ligand variants co-immunoprecipitated with NEURL1, and neither removal of the NxxN motif from DLL1 and JAG1 nor its introduction into DLL4 significantly affected the binding interaction. These results suggest that, unlike the situation in *Drosophila* where mutation of the NBM in Dl completely abolishes binding to Neur (Fontana and Posakony, 2009), mammalian NEURL1 binds Notch ligands through a more complex mechanism. Thus, although the NxxN motif is essential for ligand activation and proper localisation in mammalian cells, additional interaction sites or adapter proteins are likely involved in stabilizing the ligand-NEURL1 complex.

## Analysis of the functionality of the mammalian ICDs in Neur-dependent signalling processes in vivo

We next asked whether the ICDs of DLL1, DLL4, JAG1, and JAG2 can complement the activity of Dl at the organismal level, particularly in well-characterized Neur-dependent developmental processes in *Drosophila*. For this purpose, we generated knock-in alleles encoding hybrid ligands in which the ICD of Dl is replaced by that of the mammalian ligands (Fig. 7). We utilized the $Dl^{attP}$-landing site, as described in our previous work (Troost et al, 2023b). $Dl^{attP}$ is a null allele of Dl, causing embryonic lethality in homozygosity, because exon 6, which encodes most of Dl, is replaced by an attP landing site (Viswanathan et al, 2019). The mutant animals die as embryos due to hyperplasia of the nervous system, a neurogenic phenotype characteristic of Notch pathway mutants, including *neur* mutants (Campos-Ortega and Knust, 1990). This neurogenic phenotype is revealed by the reduction of the cuticle to a small dorsal patch in cuticle preparations (Fig. 7A, B).

We previously showed that the knock-in of the ICD of Dl into $Dl^{attP}$ resulted in a wild-type allele ($Dl^{attP}$-Dl-HA) that provided full activity and allowed correct development of the flies. In contrast,

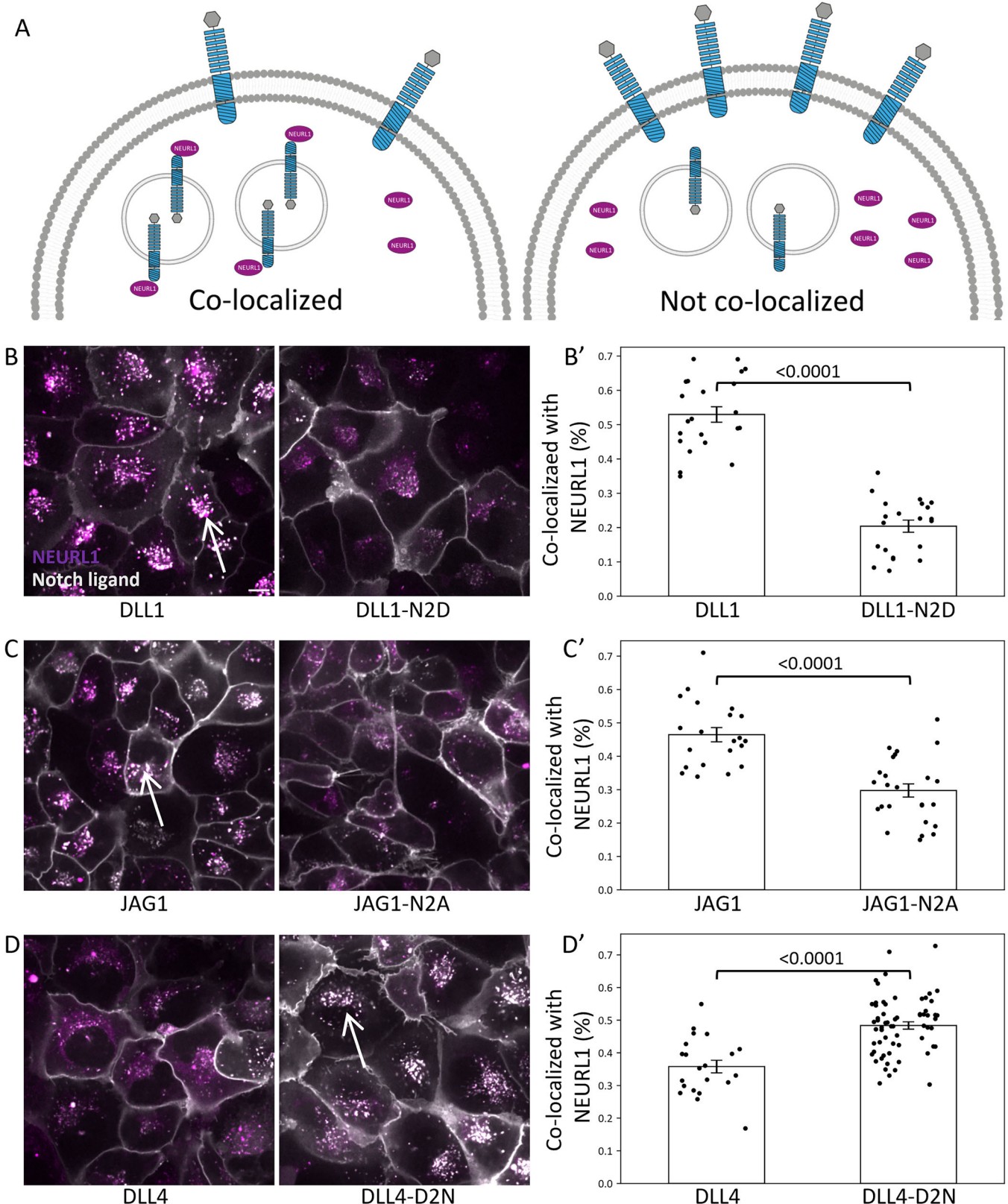

◀ **Figure 6. Reduced co-localisation between NEURL1 and Notch ligands is observed when the NxxN motif is mutated.**

(A) Co-localisation scheme. This assay measures the co-localisation ratio, which is defined by the fraction of mCherry (fused to NEURL1, shown as magenta) pixels that are co-localised with mTQ2 (fused to ligands, shown as grey) pixels. (B–D) Representative images of cells co-expressing NEURL1 fused to mCherry and one of the WT (left) or mutated (right) Notch ligands (as indicated) fused to mTQ2. Arrows mark examples of co-localisation spots. (B′, C′, D′) Quantification of co-localisation of different Notch ligands (as indicated) with NEURL1. Error bars represent the standard error. Denoted p values were calculated by the Mann–Whitney test ($n \geq 19$ images per ligand/mutant). Scale bar $= 10\,\mu m$. Source data are available online for this figure.

the knock-in of the ICD of a Dl variant with a mutated NBM failed to rescue the neurogenic phenotype in homozygosity ($Dl^{NxxN2A}$-HA, Fig. 7C, arrow) (Troost et al, 2023b).

We found that, similar to the $Dl^{attP}$-Dl-HA construct, flies homozygous for $Dl^{attP}$-DLL1-HA or $Dl^{attP}$-JAG1-HA developed to the adult stage, indicating that the ICDs of DLL1 and JAG1 provide sufficient activity for the completion of development. The correct macro- and microchaete formation on the Notum further demonstrates proper interaction with Neur during bristle development (Fig. 7E,G, compared to Fig. 7D).

In contrast, flies homozygous for $Dl^{attP}$-DLL4-HA or $Dl^{attP}$-JAG2-HA failed to rescue the Dl-mutant phenotype and died as embryos. The homozygous embryos displayed the typical neurogenic phenotype, indicating a failure of Neur-mediated activation (Fig. 7F,H, compared to Fig. 7A,B). These findings confirm that only ICDs containing an NBM can mediate Notch signalling in Neur-dependent processes and demonstrate that the NBM of DLL1 and JAG1 is fully functional in vivo.

### NEURL proteins and MIB1 are differentially expressed in different human cells and tissues

Our analysis suggests that NEURL proteins may function differently from MIB1, as they interact with only some ligands but not others. To get additional insights into the functions of NEURL proteins in vivo, we analysed expression data for MIB1, NEURL1, and NEURL1B across 54 human tissues from the Genotype-Tissue Expression (GTEx) Project database (Genotype-Tissue Expression (GTEx) Project. GTEx Analysis v10 (dbGaP accession phs000424.v10.p2, 2024). GTEx Portal. Available from: https://gtexportal.org/home/.) and single-cell RNAseq data from the Allen Brain Cell Atlas (Shen et al, 2012) (Appendix Fig. S5). We found that NEURL1 is highly expressed in specific tissues, particularly in the brain and skeletal muscle (Appendix Fig. S5A). In contrast, Mib1 and NEURL1B are more widely expressed across many tissues but at lower levels in the same tissues where NEURL1 is highly expressed (Appendix Fig. S5A, highlighted area). To take a closer look at the expression patterns within individual human neurons, we analysed publicly available single-cell RNAseq data from the Allen Brain Atlas, which confirmed similar trends. Uniform Manifold Approximation and Projection (UMAP) analysis identified 36 clusters corresponding to different neurons with different expression patterns (referred to as cell types, Appendix Fig. S5B). Most (but not all) of these cell types express high levels of NEURL1 and low levels of both MIB1 and NEURL1B (Appendix Fig. S5C–F), consistent with the tissue-level expression patterns observed in (Appendix Fig. S5A). Quantitative analysis of NEURL1, MIB1, NEURL1B, and the various Notch ligands in these cells revealed that DLL1 and JAG2 are expressed in most cell types, while JAG1 is found in some, and DLL4 is almost not expressed

across all cell types. Overall, this analysis highlights differential expression patterns of NEURL proteins and MIB1 in tissues, with NEURL1 being more highly expressed in the brain, alongside lower expression levels of MIB1 and NEURL1B. Notably, high levels of NEURL1 expression are often correlated with DLL1 expression in many cell types, and in some instances, with JAG1 expression. These ligands can be activated by NEURL1. Interestingly, these same cell types also express JAG2, which cannot be activated by NEURL1, suggesting that other activation mechanisms might be at play.

## Discussion

During *Drosophila* development, Neur-induced Notch signalling plays a crucial role in establishing the nervous system by selecting neuroblasts of the central nervous system and neural precursor cells of the peripheral nervous system from an equivalence group (Campos-Ortega and Knust, 1990; Troost et al, 2015; Corson et al, 2017; Miller and Posakony, 2018; Troost et al, 2023a). In contrast, Neurl1 and Neurl1B double mutant mice develop normally without detectable developmental defects, neuronal or otherwise, indicating that the developmental roles of Neur have been lost in the transition between flies and mammals (Koo et al, 2007). It appears that, in mammals, the developmental role of Neur has been taken over by MIB1, and it is unclear whether NEURL proteins are involved in the Notch signalling pathway at all. However, NEURL proteins are present in the adult mouse brain, where they are involved in neuronal plasticity and spatial memory, processes also dependent on Notch signalling (Yoon and Gaiano, 2005; Lee et al, 2020). Moreover, cell culture experiments show that NEURL1 can ubiquitylate JAG1 and NEURL1B can ubiquitylate DLL1, suggesting that NEURL proteins can bind to and ubiquitylate Notch ligands (Song et al, 2006; Koutelou et al, 2008). Interestingly, these in vitro experiments also suggest that the role of NEURL-mediated ubiquitylation in vertebrates is opposite to that in *Drosophila*, as it appears to downregulate signalling by JAG1 in mammals and XDelta1 in *Xenopus* (Deblandre et al, 2001; Song et al, 2006; Koutelou et al, 2008).

Here, we demonstrate, both in vivo (humanized *Drosophila*) and in mammalian cell culture, that Neurl1 (NEURL1 in cell culture) and NEURL1B can differentially activate DLL1 and JAG1, but not DLL4 and JAG2, similar to Neur in *Drosophila*. This strongly suggests that both NEURL proteins are activators of Notch signalling, analogous to their *Drosophila* counterpart Neur. Activation of ligands by NEURL proteins depends on the presence of an NxxN motif within the N-terminal region of the ligands' ICDs. This consensus sequence, previously identified as the NBM in *Drosophila*, is present in all investigated mammalian DLL1 and JAG1 homologs, but notably absent in DLL4 and JAG2 homologs

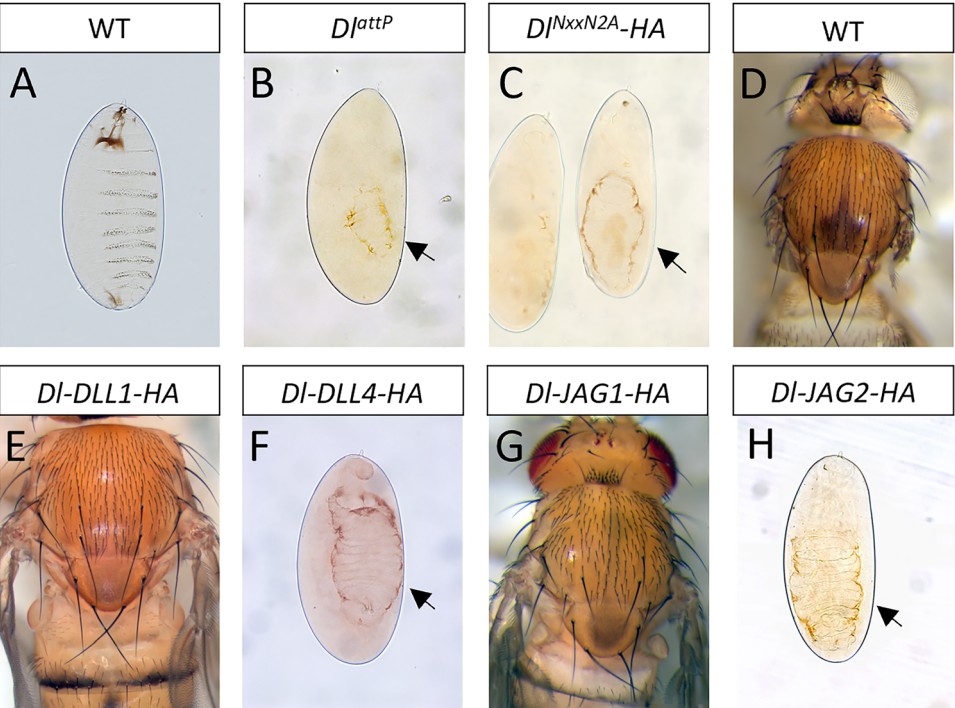

**Figure 7. In vivo analysis of the ICDs of the mammalian ligands in *Drosophila*.**

Knock-in alleles were generated that expressed the hybrid ligands instead of Dl. (A) Cuticle preparation of WT embryos at stage 17. (B) Dl$^{attP}$ embryos with neurogenic phenotype and reduced cuticle (arrow). (C) Dl$^{NxxN2A}$, a Dl allele where the NBM is mutated to alanine, showing a similar neurogenic phenotype (arrow). (D) Wildtypic Notum showing Macro- and Microchaete distribution. (E, G) Only the Dl-DLL1 and Dl-JAG1 flies develop to adulthood and show wild-type bristle formation on the notum. (F, H) Dl-DLL4 and Dl-JAG2, which do not possess an NBM, die as embryos and display the typical neurogenic cuticle phenotype (arrow). Source data are available online for this figure.

(Fontana and Posakony, 2009). The same pattern was also observed in orthologs from *Xenopus laevis* (Fig. EV1). Indeed, we showed that both rat Dll1 and human DLL1 can be activated by both Neurl proteins. Importantly, mutation of a single N residue in the NxxN motif of DLL1, rDll1, or JAG1 abolishes the ability of Neur, Neurl1/NEURL1, and NEURL1B to activate these ligands in both *Drosophila* and mammalian cell culture assays. Similarly, previous studies have shown that mutation of the NxxN motif in Dl results in loss of Neur-mediated ligand activity (Troost et al, 2023b). Furthermore, knock-in experiments replacing the ICDs of mammalian ligands in the *Dl* locus showed that only ligands containing an intact NxxN motif, DLL1 and JAG1, could prevent the neurogenic phenotype associated with loss of neur function. Thus, our findings confirm that the NxxN motif in DLL1 and JAG1 functions as an NBM in vivo.

An interesting evolutionary aspect is the presence of a cryptic NBM in mammalian Dll4, where the C-terminal N is replaced by a D. This replacement of N to an acidic D in DLL1 abolished its activation by NEURL proteins. Our experiments indicate that DLL4 can be activated in Mib1-dependent processes as efficiently as DLL1, suggesting that, unlike DLL1 and JAG1, it is devoted to Mib1-dependent signalling. However, replacing D by N in the cryptic NBM of DLL4 enabled its activation by NEURLs, indicating that the presence of the NxxN motif is sufficient for NEURL-dependent activation. The responsiveness of DLL4 to NEURLs after restoration of the cryptic site, suggests that DLL4 originally possessed a functional NBM, which was subsequently

lost during evolution, specializing DLL4 activation specifically to Mib1. Our results also indicate that JAG2 likely possesses a cryptic NBM, as it contains a similar motif in which one of the conserved N residues is replaced by the acidic amino acid, E. Notably, this substitution is fully conserved among the mammalian JAG2 orthologs examined (Fig. EV1).

We note that our co-IP analysis showed that the NxxN motif is not essential for binding between NEURL1 and mammalian Notch ligands. This result aligns with previous findings describing interactions between mouse Neurl1 and human NEURL1B with Dll1 and Dll4 in HEK293A cells (Rullinkov et al, 2009). Interestingly, pull-down experiments between the *Drosophila* Neur and the ICD of Delta, which assays direct binding, demonstrated the crucial importance of the NxxN motif for their interaction (Fontana and Posakony, 2009). Several potential explanations could account for the differences between the pull-down and Co-IP experiments. Mammalian NEURL proteins may possess additional binding sites beyond the NxxN motif. Co-IP interactions might also be indirect, mediated by additional factors, or there might be intrinsic differences in binding requirements between *Drosophila* Neur and mammalian NEURL proteins. Nevertheless, our data clearly indicate that the NBM is essential for activation of ligand-dependent signalling by Neur family proteins and for proper subcellular localisation of NEURL proteins.

Some reports suggested a role of NEURL proteins as suppressors of Notch signalling in vertebrates (Lai et al, 2001; Song et al, 2006;

Koutelou et al, 2008). A potential explanation for these contradictory reports could be the concentration-dependent manner in which Neur acts as an activator. Since Neur family members are potent inducers of endocytosis of DSL ligands (Fig. 6), maintaining a balance between Dl levels at the cell surface and the efficiency of endocytosis required for ligand activation is crucial. This balance likely depends on Neur expression levels, as suggested by overexpression experiments in *Drosophila* (Miller and Posakony, 2018). Thus, strong overexpression of NEURL1 or -1B might accelerate ligand endocytosis, reducing surface ligand levels availability and subsequent signalling, especially when MIB1 is likely already present in the cells and tissue used for the experiments. Understanding the competitive dynamics between MIB1 and NEURL1 will require carefully controlled titration experiments, and represents an intriguing area for further comprehensive investigation.

It is interesting to note that the heterozygous mutants of Jag1 and Notch1, as well as concomitant loss of Neurl1 and -1B, all cause spatial memory deficits in adult mice (Lee et al, 2020; Sargin et al, 2013). Our results, linking NEURL activity to specific ligands, could suggest a NEURL-JAG1 axis that operates during spatial memory formation and cognitive functions. Supporting this idea, our gene expression analysis shows that NEURL1 is selectively enriched in specific human brain cell types and tissues, whereas MIB1 and NEURL1B are more broadly expressed but at lower levels in the same cells and tissues. Many of these NEURL1-expressing cell types also express DLL1, and in some cases JAG1, thereby establishing in principle the necessary conditions for NEURL1-mediated Notch ligand activation. Additional experiments are required in order to directly link NEURL-dependent signalling activity to cognitive functions in mammals. Interestingly, Neur also appears to be involved in long-term memory formation in adult flies, suggesting that the post-embryonic function of NEURL family members in the brain is evolutionarily conserved and may reflect an ancestral function (Pavlopoulos et al, 2008).

While the data presented here offers strong support for the requirement of the NxxN motif, there are limitations to consider. Most of our experiments rely on overexpression systems rather than manipulation of endogenous genes, which may not fully reflect physiological conditions. However, two aspects of the experimental design help address this concern. First, activation is assessed in the mib1 mutant background in both wing disc and cell culture, where observed signalling represents a rescue of lost function rather than ectopic activity. Second, in the Delta mutant *Drosophila* strain, characterized by homozygous embryonic lethality and a neurogenic phenotype, rescue was only achieved with knock-in constructs containing the NxxN motif. These results strongly support the biological relevance of the findings.

Our study shows that all Dl-hybrids can activate the Notch pathway in *Drosophila* imaginal disks, where Mib1 is the sole E3 ligase present, except for late-occurring single sensory organ precursor cells. This indicates that, similar to mammalian MIB1, *Drosophila* Mib1 can productively interact with the ICDs of all functional mammalian DSL ligands. Thus, the mechanism of Mib1-mediated activation of the Notch pathway is conserved, and its mammalian version can be studied in these "humanized *Drosophila*", where countless techniques allow detailed analysis.

Overall, our results support a previously unidentified regulatory mechanism for differential activation of Notch signalling in mammals, based on the differential activation of Notch ligands. Hence, despite the fact that all Notch ligands can interact with all Notch receptors, the activity is restricted to specific ligands based on post-translational modifications, such as ubiquitylation, in the ligands' ICD. It will be interesting to check if this mechanism is relevant for other signalling pathways that involve combinatorial interactions between receptors and ligands.

# Methods

**Reagents and tools table**

| Reagent/resource | Reference or source | Identifier or catalogue number |
|---|---|---|
| **Experimental models** | | |
| HEK293T cells (*H. sapiens*) | ATCC | CRL-3216 |
| U2OS cells (*H. sapiens*) | ATCC | HTB-96 |
| *Drosophila melanogaster* starins (*D. mel.*): | | |
| mib1EY09870 | Lai EC, 2005 | |
| ptc-GAL4 | Speicher SA, 1994 | |
| DlattP | Viswanathan R, 2019 | |
| DlattP-Dl-NEQN2A-HA | Troost T, 2023 | |
| **Antibodies** | | |
| Anti-Flag for IP | Sigma-Aldrich | F1804 |
| Anti-Flag for WB | BioLegend | 902401 |
| Anti-HA for WB | Covance | PRB-101C |
| anti-Wg for IF | DSHB | 4D4 |
| anti-HA for IF | Cell Signalling | C29F4 |
| anti-V5 for IF | Invitrogen | #R960-25 |
| anti-myc for IF | Cell Signalling | 9B11 |
| anti-rabbit Alexa568 for IF | Invitrogen | A-21235 |
| anti-mouse Alexa647 for IF | Invitrogen | A-11011 |
| **Chemicals, enzymes and other reagents** | | |
| Lipofectamine 3000 | Thermo Fisher | L3000008 |
| TransIT-X2 | Mirus | MIR 6000 |
| DMEM, high glucose with L-glutamine | Sartorius | 01-052-1 A |
| Penicillin streptomycin | Biowest | L0022-100 |
| Fetal bovine serum (FBS) | Biowest | S1400-500 |
| Protease inhibitor cocktail | Calbiochem | 539131 |
| Anti-mouse IgG-agarose beads | Sigma-Aldrich, Merck | A-6531 |
| PFA | Electron Microscopy Sciences | 19200 |
| Triton X-100 | Sigma-Aldrich, Merck | 1003267647 |
| Normal goat serum (NGS) | Biozol Diagnostica Vertrieb GmbH | ENG9010-10 |

| Reagent/resource | Reference or source | Identifier or catalogue number |
|---|---|---|
| Vectashield | Vector laboratories | H-1000 |
| Passive lysis buffer | Promega | E1941 |
| Wizard(R) Plus SV Minipreps DNA Purif. | Promega | A1460 |
| Wizard(R) SV gel and PCR clean-up system | Promega | A9282 |
| KAPA HiFi HS | Roche | 7958935001 |
| Gibson Assembly Master Mix | NEB | NEB-E2611S |
| **Software** | | |
| Imagej Fiji | https://imagej.net/ij/ | |
| Python | https://www.python.org/ | |
| Adobe Illustrator 2024 | https://www.adobe.com/il_en/products/illustrator.html | |
| Mann–Whitney U-calculator | https://www.socscistatistics.com/tests/mannwhitney/ | |
| Adobe Photoshop | https://www.adobe.com/de/products/photoshop.html | |
| Qiagen CLC Main Workbench | https://digitalinsights.qiagen.com/trial-request/ | |
| Biorender | https://www.biorender.com/ | |
| Benchling | https://www.benchling.com/ | |

## Fly stocks

$mib1^{EY09870}$ (Lai EC, 2005), $ptc$GAL4 (Speicher SA, 1994), $Dl^{attP}$ (Viswanathan R, 2019), $Dl^{attP}$-$Dl$-NEQN2A-HA (Troost, 2023); $Dl^{attP}$-hDLL1-HA, $Dl^{attP}$-hDLL4-HA, $Dl^{attP}$-hJAG1-HA, $Dl^{attP}$-hJAG2-HA, UAS-Dl-rDll1-HA, UAS-Dl-rDll1-N587A-HA, UAS-Dl-rDll1-N587D-HA, UAS-Dl-rDll1-NBM2A-HA, UAS-Dl-hDLL1-HA, UAS-Dl-hDll1-N595D-HA, UAS-Dl-hDLL4-HA, UAS-Dl-hDLL4-D573N-HA, UAS-Dl-hJAG1-HA, UAS-Dl-hJAG1-N1110A-HA, UAS-Dl-hJAG2-HA, UAS-mNeurl1-myc, UAS-hNEURL2-myc, and UAS-Dneur-V5 (this study).

## Antibody staining of wing imaginal disks

Wing imaginal disks of wandering L3 larvae were dissected in PBS, fixed with 4% paraformaldehyde in PBS (4%PFA) for 30 min and washed with 0.3% Triton X-100 in PBS (PBT). Permeabilization and blocking was done using a 5% normal goat serum (NGS) in PBT for 1 h at room temperature. Primary antibody incubation was performed in 5% NGS in 0.3% PBT for 2 h at room temperature, followed by three washing steps with PBT. The corresponding secondary antibody was applied in 5% NGS in 0.3% PBT for 2 h at room temperature. The following antibodies were used: mouse anti-Wg 4D4 (1:10, Developmental Studies Hybridoma Bank (DSHB), Iowa City, IA, USA), rabbit anti-HA (Cell Signalling C29F4), mouse anti-V5 (Invitrogen #R960-25), and mouse anti-myc (Cell Signalling 9B11). Fluorophore-conjugated secondary antibodies were purchased from Invitrogen.

Nuclei staining was done using the Hoechst 33258 dye. For further information, see Klein (Klein, 2008).

## Generation of DNA constructs

The constructs mNeurl1 and hNEURL2 were cloned from pEGFP-Vector into pUAST-attB-Vector for GAL4-driven expression in *Drosophila* via restriction cloning with the enzymes EcoRI and XhoI for mNeurl1 and EcoRI and XbaI for hNEURL2 and tagged with a C-terminal myc epitope. The constructs were inserted into the landing site 86Fb on the third chromosome and recombined with the allele $mib1^{EY09780}$. The hybrid ligands Dl-rDll1-HA, Dl-hDLL4-HA, Dl-hJAG1-HA, and Dl-hJAG2-HA were generated via overhang extension PCR and cloned into the pUAST-attB-Vector via the restriction sites NotI and XbaI and inserted into the landing site 51C on the second chromosome. The ICD of hDLL1 was ordered from IDT technologies and cloned into the pUAST-attB-Dl-HA vector via the restriction sites BstEII and XbaI. Single-point mutations were introduced via SDM. For endogenous expression under the delta promoter, the hybrid ligands were cloned into the pGE-attB-Vector with the restriction enzymes BstEII and XbaI and inserted into the *DlattP* landing site (Viswanathan et al, 2019). The list of primers used for cloning can be found below (Table 1).

Cell culture constructs were generated using the Gibson Assembly method (Gibson et al, 2009). Cloning protocols are described in (Khamaisi et al, 2022). Full plasmid maps and sequences can be found below (Table 2).

## Cells, materials, and constructs

In this study, we used HEK293T (for lentivirus production) and U2OS cell lines, which were cultured in Dulbecco's Modified Eagle's Medium (DMEM) supplemented with 10% fetal bovine serum (FBS). The cells were maintained at a constant temperature of 37 °C and an atmosphere of 5% $CO_2$. Transfection of the cells was executed using Lipofectamine 3000 (Thermo Fisher Scientific) following the manufacturer's instructions.

Sender cells were generated through lentiviral transduction. This involved employing a pLVX vector encoding one of the ligands or NEURL1, co-transfected with helper plasmids (pMD2.G and PsPax2), into HEK293T cells using Lipofectamine 3000. Following a 48-h incubation period, the supernatant was carefully collected, filtered using a 0.45 μm filter, and subsequently utilized for transduction on either U2OS WT or U2OS Mib1 KO cells.

Our receiver cells, U2OS-Notch1-Gal4 and U2OS-Notch2-Gal4 were kindly provided by Stephen C. Blacklow (Harvard Medical School) (Gordon et al, 2009).

## Luciferase activity test

The activity of NEURL1 with the different ligands was measured with a luciferase activity assay. The detailed protocol for the luciferase assay is described in (Allen Brain Cell Atlas—Data Access). Briefly, U2OS-NOTCH1-Gal4 or U2OS-NOTCH2-Gal4 (receiver cells) were cultured in a 24-well plate and transfected with 350 ng of UAS-firefly luciferase reporter (Andrawes et al, 2013) and 10 ng of pRL-SV40 Renilla luciferase. Transfection was done using Lipofectamine 3000 (Thermo Fisher Scientific). Twenty-four hours after transfection, the receiver cells were transferred to a new 24-well plate with doxycycline (200 ng/ml), co-cultured with sender

**Table 1. Primers**

| Primer | 5′ to 3′ Sequence |
|---|---|
| EcoRI-mNeurl1-for | gattggtgaattcatgggtaacaacttctccagtgt |
| mNeurl1-myc-Xhol-rev | Tacctctcgagctactaaagatcttcgctaataagtttttgttctcc tcctcccgaaccggtggagctgcggtaggtcttgatgatg |
| EcoRI-hNEURL1B-for | gattggtgaattcatgggcaacacggtgcaccggac |
| hNEURL1B-myc-XbaI-rev | Taccttctagactactaaagatcttcgctaataagttttt gttcctcctcccgaaccggttggcctgtagatcttaatgacgtcc |
| NotI-Delta-for | tgcaggcggccgcatgcattggattaaatgtttattaacagc |
| Delta-rDll1-rev | ttgcggcgtgcgtggtcttctgcgtccggctgaagcta |
| Delta-rDll1-for | tagcttcagccggacgcagaagaccacgcacgccgcaa |
| rDll1-HA-rev | gatcctctagagcggccgcactgagcagcgtaatctggaacg |
| Delta-hDLL4-rev | cgaagccgcagctgccgcacgaagaccacgcacgccgc |
| Delta-hDLL4-for | gcggcgtgcgtggtcttcgtgcggcagctgcggcttcg |
| hDll4-HA-rev | ctgcatctagattaagcgtagtctggaacgcgtatgggtacgctacctccgtggcaatgacac |
| Delta-hJAG1-rev | ctgcccggcttccgccgcttccggaagaccacgcacgccgcaataac |
| Delta-hJAG1-for | gttattgcggcgtgcgtggtcttccggaagcggcggaagccgggcag |
| hJAG1-HA-rev | gatcctctagaCTAagcgtagtctggaacgtcgtatgggtacgctacgatgtactccattcgg |
| Delta-hJAG2-rev | cccgctctttcctgcgcttgcgtgtgaagaccacgcacgccgcaataac |
| Delta-hJAG2-for | gttattgcggcgtgcgtggtcttcacacgcaagcgcaggaaagagcggg |
| hJAG2-HA-rev | Ctgcatctagactaagcgtagtctggaacgtcgtatgggtacgcctccttgccggcgtagcgggcctc |
| SDM-rDll1-N587A-for | GAACAacctagccGCCtgccagcgtgagaaggatgtttctg |
| SDM-rDll1-N587A-rev | ctcacgctggcaGGCggctaggttgttcatggtctctgtc |
| SDM-rDll1-N587D-for | GAACAacctagccGACtgccagcgtgagaaggatgtttctg |
| SDM-rDll1-N587D-rev | ctcacgctggcaGTCggctaggttgttcatggtctctgtc |
| SDM-rDll1-NBM2A-for | gagacagagaccatgaacGCCGCCGCCGCCtgccagcgtgagaaggatgtttctgttag |
| SDM-rDll1-NBM2A-rev | cacgctggcaGGCGGCGGCGGCgttcatggtctctgtctctccccccgcaaggatcagg |
| SDM-hDLL4-D573N-for | gccatgaacaacttgtcgAACttccagaaggacaacc |
| SDM-hDLL4-D573N-rev | gacaagttgttcatggcttccctgctgc |
| SDM-hJAG1-N1110A-for | Gccacacacactcagcctctgaggacgccaccaccaacaacgtgcggg |
| SDM-hJAG1-N1110A-rev | Gttggtggtggcgtcctcagaggctgagtgtgtgtggctgcccggcttccg |
| SDM-hJAG1-NBM2A-for | cagcctctgaggacGCAGCAGCAGCAaacgtgcgggagcagctgaacc |
| SDM-hJAG1-NBM2A-rev | cccgcacgttTGCTGCTGCTGCgtcctcagaggctgagtgtgtgtgg |
| SDM-hDLL1-N595D-for | catgaacaacctagccGACtgccagcgtgag |
| SDM-hDLL1-N595D-rev | gtctctgtctctccCCTgcaagg |

cells expressing NEURL1 and one of the Notch ligands. After additional 24 h the cells were lysed with passive lysis buffer (Promega, 100 μl/well) and taken to be measured with luminometer software (GloMax(R) Navigator Microplate Luminometer, Navigator 2010, Promega). Notch activity was defined as the ratio of luciferase to Renilla, which was then normalised by the negative control to represent fold change in activity.

## Microscopy

Images of wing imaginal disks were acquired with the Zeiss Axio Imager Z1 Microscope equipped with a Zeiss Apotome2.

Imaging of cells utilized an Andor revolution spinning disk confocal microscope, powered by 50 mW lasers from Andor in Belfast,

Northern Ireland. The microscope featured a 37 °C temperature-controlled chamber and a $CO_2$ regulator providing 5% $CO_2$, both supplied by Okolab in Italy. The setup comprised an Olympus inverted microscope with an oil-immersion Plan-Apochromatic 60× objective NA = 1.42 from Olympus in Tokyo, Japan, along with an ANDOR iXon Ultra EMCCD camera also from Andor in Belfast, Northern Ireland. Control of the equipment was facilitated by Andor iQ software from Andor in Belfast, Northern Ireland.

## Quantification of over-proliferation and ectopic Wg activation of wing imaginal disks

In order to quantify the trans-activation of Notch Signalling in an in vivo setting, the over-proliferation of wing imaginal disks was

**Table 2. Plasmid maps**

| Construct | Link |
|---|---|
| pLVX-Ef1a-DLL1-mTQ2 | https://benchling.com/sprinzak/f/lib_1yc7Spzl-neuralized-paper-plasmids/seq_I5gCz7kY-plvx-ef1a-dll1-mtq2/edit |
| pLVX-Ef1a-DLL1-N2D-mTQ2 | https://benchling.com/sprinzak/f/lib_1yc7Spzl-neuralized-paper-plasmids/seq_g45E5dOD-plvx-ef1a-dll1-n2d-mtq2/edit |
| pLVX-Ef1a-DLL4-mTQ2 | https://benchling.com/sprinzak/f/lib_1yc7Spzl-neuralized-paper-plasmids/seq_Dlgw9WOl-plvx-ef1a-dll4-mtq2/edit |
| pLVX-Ef1a-DLL4-D2N-mTQ2 | https://benchling.com/sprinzak/f/lib_1yc7Spzl-neuralized-paper-plasmids/seq_CyaZp2Dp-plvx-ef1a-dll4-d2n-mtq2/edit |
| pLVX-Ef1a-JAG1-mTQ2 | https://benchling.com/sprinzak/f/lib_1yc7Spzl-neuralized-paper-plasmids/seq_9rc6xBhC-plvx-ef1a-jag1-mtq2/edit |
| pLVX-Ef1a-JAG1 -N2A-mTQ2 | https://benchling.com/sprinzak/f/lib_1yc7Spzl-neuralized-paper-plasmids/seq_EiKpJaoM-plvx-ef1a-jag1-n2a-mtq2/edit |
| pLVX-Ef1a-JAG2-mTQ2 | https://benchling.com/sprinzak/f/lib_1yc7Spzl-neuralized-paper-plasmids/seq_FY0BhFj2-plvx-ef1a-jag2-mtq2/edit |
| pLVX-Ef1a-NEURL1-mCherry | https://benchling.com/sprinzak/f/lib_1yc7Spzl-neuralized-paper-plasmids/seq_oi86kpGC-plvx-ef1a-neurl1-mcherry/edit |
| pLVX-Ef1a-DLL1-mTQ2-HA | https://benchling.com/sprinzak/f/lib_1yc7Spzl-neuralized-paper-plasmids/seq_urcrtgey-plvx-ef1a-dll1-mtq2-ha/edit |
| pLVX-Ef1a-DLL1-N2D-mTQ2-HA | https://benchling.com/sprinzak/f/lib_1yc7Spzl-neuralized-paper-plasmids/seq_urcrtgey-plvx-ef1a-dll1-mtq2-ha/edit |
| pLVX-Ef1a-DLL4-mTQ2-HA | https://benchling.com/sprinzak/f/lib_1yc7Spzl-neuralized-paper-plasmids/seq_DUVSsvV4-plvx-ef1a-dll4-mtq2-ha/edit |
| pLVX-Ef1a-DLL4-D2N-mTQ2-HA | https://benchling.com/sprinzak/f/lib_1yc7Spzl-neuralized-paper-plasmids/seq_X1wNBoZh-plvx-ef1a-dll4-d2n-mtq2-ha/edit |
| pLVX-Ef1a-JAG1-mTQ2-HA | https://benchling.com/sprinzak/f/lib_1yc7Spzl-neuralized-paper-plasmids/seq_ZRwBPGRp-plvx-ef1a-jag1-mtq2-ha/edit |
| pLVX-Ef1a-JAG1 -N2A-mTQ2-HA | https://benchling.com/sprinzak/f/lib_1yc7Spzl-neuralized-paper-plasmids/seq_K11TARdC-plvx-ef1a-jag1-n2a-mtq2-ha/edit |
| pLVX-Ef1a-JAG2-mTQ2-HA | https://benchling.com/sprinzak/f/lib_1yc7Spzl-neuralized-paper-plasmids/seq_MEmSPemk-plvx-ef1a-jag2-mtq2-ha/edit |
| pLVX-Ef1a-NEURL1-mCherry-FLAG | https://benchling.com/sprinzak/f/lib_1yc7Spzl-neuralized-paper-plasmids/seq_aeqWcCJ0-plvx-ef1a-neurl1-mcherry-flag/edit |

measured. Using the software ImageJ, the length of the anterio-posterior boundary was measured and divided by the length of the dorso-ventral boundary (A/P:D/V). The result shows the fold change in proliferation induced by ectopic activation of Notch-Signalling. For each genotype, five wing imaginal disks were measured. For statistical analysis, a Mann–Whitney $U$-test was applied.

For the measurement of ectopic Wg activation, the length of cells ectopically expressing Wg was measured on both sides of the ptc-expression domain. The measured pixels were then normalised to the length of the D/V boundary.

## Co-localisation assay

The co-localisation assay was performed using live imaging microscopy with cells co-expressing NEURL1 (fused to mCherry) and a DSL-ligand (fused to mTQ2). Snaps were taken in two channels, red and blue (561 and 445 nm, respectively). All images were analysed using custom code in Python (https://github.com/Oren-Gozlan/Co-localization-code) to calculate the co-localisation ratio. This was performed by taking all the pixels that have both red and blue values (above a threshold) and dividing them by all the red pixels (above the same red threshold). Each DSL-ligand was compared to its NBM-related variant (e.g., DLL1 vs DLL1-N2D) and significance between samples was calculated with a Mann–Whitney $U$-test.

## IP and immunoblotting

For the co-IP experiment, 1 mg of the lysed cells [50 mM Hepes (pH 7.5), 3 mM EDTA, 3 mM CaCl$_2$, 80 mM NaCl, and 1% Triton] supplemented with protease inhibitor cocktail (Calbiochem

#539131) and 1 nM DTT were incubated at 4 °C for 1 h. After removal of cellular debris by centrifugation (10 min × 20,000 × $g$), lysates were incubated with 50% anti-mouse IgG-agarose beads (Sigma-Aldrich, Merck # A-6531) together with 10 µg/ml mouse anti-FLAG M2 Ab (Sigma #F1804) for 18 h at 4 °C. The immunoprecipitated proteins or total extracts (100 µg protein) from each treatment were subjected to 4–15% Criterion TGX Precast Gradient Gel (Bio-Rad Laboratories, CA, USA) and electroblotted (45 min, 100 V) onto nitrocellulose membranes in the presence of blotting buffer (186 mM glycine, 25 mM Tris base, and 20% methanol). Uniformity of sample loading was verified by Ponceau staining of the blots. Each blot was blocked for 1 h in 10 mM Tris base, 150 mM NaCl containing 5% fat-free milk, then incubated for 16 h at 4 °C with the primary Ab: Rabbit polyclonal anti-HA.11 Ab (COVANCE #PRB-101C) or rabbit anti-FLAG Ab (Biolegend # 902401). Rabbit IgG peroxidase conjugates (1:10,000) were used as a second antibodies. The blots were developed using the Immobilon Crescendo Western HRP Substrate (Millipore, Merck).

## RNA expression across human tissues

The Genotype-Tissue Expression (GTEx) project (v10) has analysed gene expression in over 19,000 samples across 54 tissues from over 1000 individuals (Genotype-Tissue Expression (GTEx) Project. GTEx Analysis v10 (dbGaP accession phs000424.v10.p2, 2024). GTEx Portal. Available from: https://gtexportal.org/home/.). The data used for the analyses described in this manuscript were obtained from the GTEx Portal on 06/17/2025 and dbGaP accession number phs000424.v10.p2 on 06/17/2025.

Visualization of the NEURL1, NEURL1B, and MIB1 expression across tissues were performed using Matplotlib (Hunter, 2007),

Seaborn (Waskom, 2021), and Pandas (The pandas development team, 2025) Python libraries.

## Single-cell RNA-seq analyses of human neuronal gene expression

Single-cell RNA gene expression was analysed using data from the Allen Brain Cell Atlas, focusing on the Whole Human Brain 10x v3 neuronal dataset (WHB-10X v3-Neurons-raw.h5ad, released 01/26/2024) downloaded on 06/23/2025. This dataset comprises ~2,480,956 cells, covering about 36,601 genes across diverse neuronal populations (Allen Brain Cell Atlas—Data Access) The raw data was downloaded from Allen Institute's AWS repository (https://allen-brain-cell-atlas.s3.us-west-2.amazonaws.com/index.html#expression_matrices/WHB-10Xv3/20240330/). Preprocessing and normalisation were performed using the Scanpy Python package (Wolf et al, 2018), including library size normalisation (library size scaling to 10,000), log-transformation, selection of 2000 highly variable genes, scaling, principal component analysis (PCA), and Leiden clustering for defining distinct cell populations. UMAP dimensionality reduction and cluster visualizations were performed using Scanpy as well as Matplotlib (Hunter, 2007) and Pandas (The pandas development team, 2025). Differential expression analysis to identify cluster-specific marker genes and visualization of NEURL1, NEURL1B, MIB1, DLL1, DLL4, JAG1, and JAG2 expression were conducted using Scanpy plot functions.

## Data availability

The datasets produced in this study are available in the following databases: All data for quantitative figures (Figures + analysis) has been uploaded to the source data files. Images used for quantification: Zenodo database, https://doi.org/10.5281/zenodo.15861040. RNA-seq expression data: Genotype-Tissue Expression (GTEx) Project, accession number phs000424.v10.p2.

The source data of this paper are collected in the following database record: biostudies:S-SCDT-10_1038-S44319-025-00601-7.

## Peer review information

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

## Acknowledgements

We thank Stefan Kölzer for excellent support in the generation of the described transgenic flies and Ilka Wolff and Manuel Sopar for excellent assistance. We thank F. Schweisguth and G. Boullianne for providing fly stocks. We thank Stephen C. Blacklow for providing the Notch-Gal4 and MIB1-KO cell lines. We thank Geraldine A. Weinmaster for the Neurl constructs. This work was supported by a Middle-East Grant KL 1028/13-1 of the Deutsche Forschungsgemeinschaft (DFG), awarded to TK and DS.

## Author contributions

**Alina Airich**: Conceptualization; Formal analysis; Investigation; Visualization; Writing—original draft; Writing—review and editing. **Oren Gozlan**: Conceptualization; Formal analysis; Investigation; Visualization; Writing—original draft; Writing—review and editing. **Ekaterina Seib**: Investigation. **Gittel Leah Shaingarten**: Conceptualization; Formal analysis; Investigation; Visualization; Writing—original draft; Writing—review and editing. **Lena-Sophie Wilschrey**: Investigation. **Liora Lindenboim**: Conceptualization; Supervision; Funding acquisition; Investigation; Visualization; Writing—original draft; Writing —review and editing. **David Sprinzak**: Conceptualization; Supervision; Funding acquisition; Investigation; Writing—original draft; Writing—review and editing. **Thomas Klein**: Conceptualization; Supervision; Funding acquisition; Investigation; Visualization; Writing—original draft; Writing—review and editing.

Source data underlying figure panels in this paper may have individual authorship assigned. Where available, figure panel/source data authorship is listed in the following database record: biostudies:S-SCDT-10_1038-S44319-025-00601-7.

## Funding

## Disclosure and competing interests statement

The authors declare no competing interests.

# Expanded View Figures

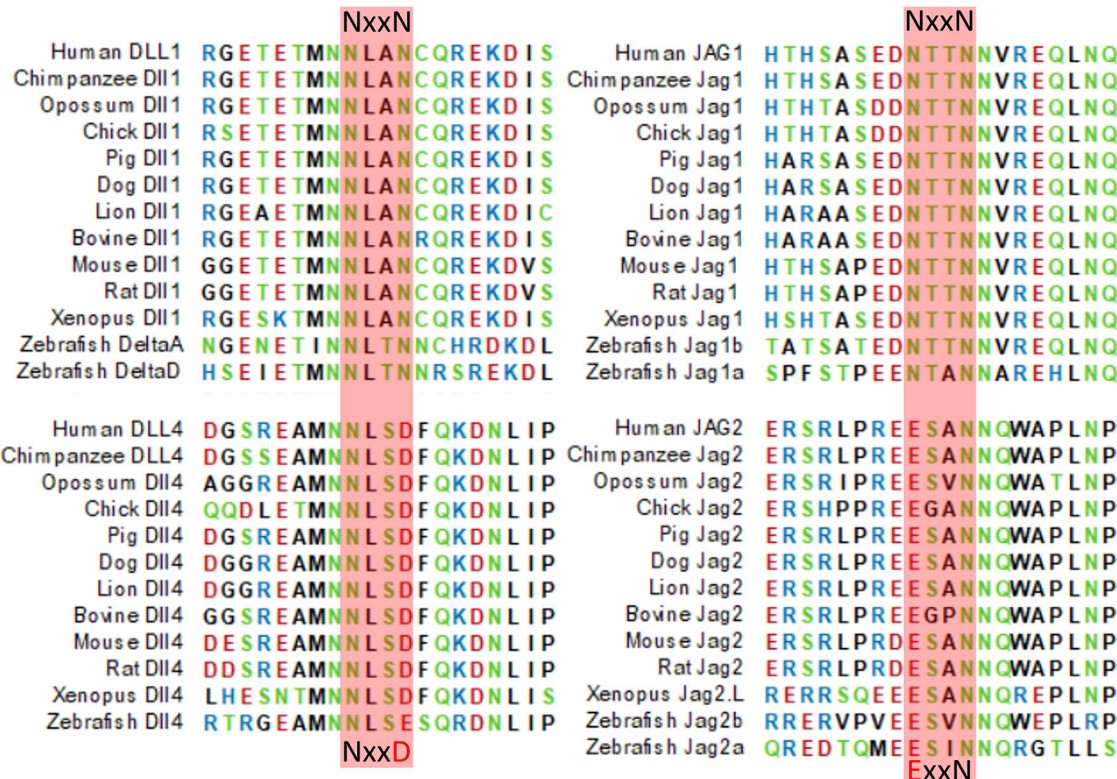

**Figure EV1.  Partial sequence comparison of Notch ligands.**

A potential NBM is present in model organisms such as *Xenopus*, *Zebrafish* and *Mouse*, as well as several other species. In DLL1 and JAG1, a conserved NxxN motif is present, while in DLL4 and JAG2, a cryptic motif is present, which contains a D/E instead of an N. These motifs are all located at the N-terminus of the ICDs close to the plasma membrane

