## [Peer Review File · EMBO Reports]

Neutralized-like proteins differentially activate Notch ligands

Alina Airich, Oren Gozlan, Ekaterina Seib, Gittel Shaingarten, Lena-Sophie Wilschrey, Liora Lindenboim, David Sprinzak, and Thomas Klein

Corresponding author(s): David Sprinzak (davidsp@tauex.tau.ac.il), Thomas Klein (thomas.klein@hhu.de)

Review Timeline:

Submission Date:	19th Dec 24
Editorial Decision:	10th Jan 25
Revision Received:	30th Jun 25
Editorial Decision:	21st Aug 25
Revision Received:	12th Sep 25
Accepted:	29th Sep 25

Editor: Deniz Senyilmaz Tiebe / Esther Schnapp

**Transaction Report: This manuscript was transferred to
EMBO reports following peer review at Review Commons.**

**Review
COMMONS**

Review #1

1. Evidence, reproducibility and clarity:

Evidence, reproducibility and clarity (Required)

This is an interesting manuscript from two groups of experts in Notch signaling biology with complementary expertise in *Drosophila* genetics (Klein) and in biophysical studies of the Notch pathway (Sprinzak). The paper provides a cutting-edge structure-function dissection of the E3 ubiquitin ligase Neuralized and its mammalian homologs, Neurl1a and Neurl1b. The work is particularly relevant since the functions of mammalian Neurl1a and Neurl1b have been questioned, and more subtle altogether than those of fly Neuralized (as summarized by the authors in Fig. 1C). This is in part due to the dominant effects of the E3 ubiquitin ligase Mindbomb1 (Mib1) in Notch ligand-expressing cells from mammalian systems. The authors use careful structure-function work in fly development (mostly wing imaginal discs) and in mammalian cell culture systems, including a clever approach to study the function of mammalian Neurl1a and Neurl1b and mammalian/fly Notch ligand hybrids in *Drosophila* to draw new conclusions about the function of Neurl1a/b, showing that they can function as activators of Notch signaling mediated by the Notch ligands Dll1 and Jag1, and not by Dll4 and Jag2, tracing these differential effects to the recognition of a short NXXN consensus sequence in the N-terminal region of the ligand's intracellular domain.

Specific questions:

- The current title of the manuscript is not very information-rich and would not allow a reader to gather key information about the findings without reading at least the abstract. Could this be improved? For example, by referring to differential activation of individual Notch ligands, or some other more direct description of the key findings?
- The authors design most key experiments documenting agonistic effects of Neurl1a/1b in a Mib1-deficient background, both in flies and in cell culture systems. This is understandable experimentally to isolate Neurl1a/b's effects in these experimental systems. However, this leaves open questions as to the prevailing effects of Neurl1a/b in cells that also express Mib1 (which the authors comment on in the discussion based on past findings, including some suggesting that Neurl1a/1b can function as Notch inhibitors through a ligand ubiquitination mechanism that may differ from their activating function). Do the authors actually have data that could shed light on this discussion? For example, have they performed cell coculture assays in which Neurl1a or Neurl1b is co-expressed with a Notch ligand, but in the presence of Mib1? This condition seems to be systematically

omitted from all the coculture experiments that are presented. It would be interesting to evaluate the net effect of Neurl1a/Neurl1b expression in a Mib1-sufficient system as well.

- The paper suggests important predictions about mammalian functions of Neurl1a/1b, including the neurological effects that have been reported, in double-deficient mice, namely that there are cells that only express Neurl1a/1b and not Mib1 and do rely on Dll1 and Jag1 for signaling. Could the authors at least comment on this prediction? Are there any single cell atlases where candidate cells like that can be identified? Or would the authors predict that Neurl1a/1b could actually function as Notch agonist even in cells expressing Mib1? (see also previous comment)

- Some minor typos: line 305 should likely read "flies homozygous for (...)". Line 408, "for providing" repeated twice.

2. Significance:

Significance (Required)

Thank you for the opportunity to review this lovely collaborative paper. As indicated in my comments to the authors, the findings provide novel structure-function information about an understudied aspect of Notch signaling and clarify conflicting past data about the mammalian homologs of fly Neuralized. The approach is elegant and multidisciplinary, notably in regards to the combination of cell co-culture systems and *Drosophila* as a platform to study mammalian Neuralized proteins and hybrid Notch ligand molecules. The findings will be interesting to the field and will generate discussion. I would suggest that some additional information would be a plus to substantiate predictions about mammalian functions of Neurl1a/b, and also to clarify its effects in the presence or absence of concomitant Mib1 expression.

3. How much time do you estimate the authors will need to complete the suggested revisions:

Estimated time to Complete Revisions (Required)

(Decision Recommendation)

Between 1 and 3 months

4. Review Commons values the work of reviewers and encourages them to get credit for their work. Select 'Yes' below to register your reviewing activity at Web of Science Reviewer Recognition Service (formerly Publons); note that the content of your review will not be visible on Web of Science.

No

Review #2

1. Evidence, reproducibility and clarity:

Evidence, reproducibility and clarity (Required)

****Summary****

The manuscript describes an analysis of specificity of functional interactions between mammalian Neuralized proteins and different human ligands for Notch. To investigate this, the authors take the approach of constructing hybrid proteins that contain the intracellular domain of the human ligands and the extracellular domain of the *Drosophila* Delta or Serrate, and investigate their activity *in vivo*, in the *Drosophila* wing disc. The latter is a well-established model tissue for assessing Notch ligand activity. As a second assay they express mammalian neutralized constructs in human cells for luciferase-based Notch signal reporter assays. The experiments are well presented and described and make a strong case for the conclusions that both Neurl1 and 2 can activate Notch signalling by Dll1 and Jag1 but not Dll4 and Jag2. Use of different mutant intracellular domains is used to show the importance of the NXXN motif, which in *Drosophila* is required for Neuralized interaction with Delta and Serrate. The use of missense mutations and in particular the reactivation of the cryptic NXXD site in Dll4 by substitution to N is convincing for establishing the importance of the motif. There is also colocalization data to support the conclusion that there is likely to be NXXN-dependent complex formation between the ligand and Neuralized proteins. This latter conclusion would be made firmer if there were pull down data to support it, although to be fair it is most unlikely that another explanation, other than complex formation could account for the observation of both colocalization and ligand activation.

****Major comments****

The main limitation of the work is that it is mostly based on overexpression of constructs to activate ectopic expression rather than gene editing endogenous genes. It would be helpful if the authors could comment on the limitations of the work in discussion. Two points of data included in the work are important in mitigating this limitation. Firstly, the experiments

in the wing disc and cell culture are taking place in a mindbomb mutant background and the activation is observed is therefore a rescue of activity that has been lost. Secondly, and importantly, the final experiment makes use of a Dl mutant Drosophila line which shows embryo lethality when homozygous, with the characteristic neurogenic phenotype. Rescue of lethality can be brought about by knock-in experiments which restore Dl function and this is also true for the ligand hybrid constructs that introduce mammalian ligand intracellular domains only when they include the NXXN motif This indicates the importance of the motif in normal development

Overall, the data presented in the paper is convincing as regards the conclusions made.

****Minor points****

In figure 1 the legend for D says that cryptic sites are substitutions of N for E or Q, but the figure and main text indicate that the substitutions are N to E or D.

In the remain figures it would be helpful to include in the figure legends and indications of the numbers of wing discs, embryos for which the images shown are representative of.

In Fg 3 The activation of Notch, by neural1 and DL-Jag1 in B''' is stronger in the ventral side of the disc than the dorsal whereas, although activation of the same ligand by Neurl2 in C''' is weaker the majority of the ectopic wingless expression is on the dorsal compartment. Is there any reason for the switch in preference between the two neutralized proteins? Overgrowth of the wing disc seems to be similar on both sides and so am wondering if the picture is representative of the ectopic wingless distribution in this case.

2. Significance:

Significance (Required)

Previous work on double genetic knockouts of the two mouse Neuralized genes cast doubt as to whether Neuralized proteins play a role in Notch signal activation in mammals, unlike in Drosophila. There is, however, some genetic indications that spatial memory requires both Notch and neuralized proteins and may represent a specialised function limited to the Neuralized interaction. There are likely to be more subtle contexts waiting to be uncovered. The work is therefore showing important proof of principle for establishing the functionality of the mammalian Neurl proteins and highlights new findings indicting specialisation of the different ligands for interactions with Notch components. Elucidation of such specialisations will help understand why the diversity of different homologues of

Notch and ligand have evolved and are maintained in the vertebrate genome compared to the single Notch and two ligands in *Drosophila*. Since Notch and its misregulation are widely involved in development, health and disease and there is much interest in developing therapeutic interventions that alter Notch activity then the work is likely of broad interest.

3. How much time do you estimate the authors will need to complete the suggested revisions:

Estimated time to Complete Revisions (Required)

(Decision Recommendation)

Less than 1 month

4. Review Commons values the work of reviewers and encourages them to get credit for their work. Select 'Yes' below to register your reviewing activity at Web of Science Reviewer Recognition Service (formerly Publons); note that the content of your review will not be visible on Web of Science.

Yes

Review #3

1. Evidence, reproducibility and clarity:

Evidence, reproducibility and clarity (Required)

****Summary****

Notch signalling is one of the major evolutionarily conserved signalling pathways involved in numerous developmental, physiological and pathological processes. Activation of the Notch receptor first requires ubiquitination of its ligands (collectively termed DSL), leading to a 'pulling force' that, upon ligand-receptor engagement, exposes Notch to intramembrane proteolysis leading to the nuclear translocation of the receptor's intracellular domain and activation of target genes with its DNA-binding co-factors.

While both Neuralized (Neurl) and Mind bomb are the E3 ubiquitin ligases for Notch ligands required for *Drosophila* development, in mammals, the Neur homologues Neur1 (officially

Neur1) and Neur2 (officially Neur1B) are dispensable for development since double Neur1/2 knock-out mice have no developmental phenotype (but both Neur homologues are involved in the memory-related functions of Notch pathway in adulthood). Rather, just one of the two mammalian Mind bomb homologues, Mib1, functions as the chief E3 ligase for Notch ligands during mammalian development as evidenced by its Notch-related knockout phenotype.

Therefore, it has not been fully established whether and how the NEUR proteins regulate the mammalian Notch ligands.

To clarify this issue, the authors assessed the capability of *Drosophila* Neur and mammalian NEUR1 and 2 proteins to activate the various hybrid Notch ligands (containing extracellularly *Drosophila* Delta and intracellularly the various Notch ligands' intracellular domains) in *Drosophila* wing discs and mammalian cell culture. The authors found that NEUR proteins only activate the Notch ligands containing a Neuralized binding motif, with the consensus sequence NxxN, that is present in DLL1 and JAG1, but not in DLL4 and JAG2. The authors also analyse the intracellular domains of mammalian Notch ligands DLL1, DLL4, JAG1 and JAG2 in *Drosophila* by generating knock-in alleles where endogenous DI expression had been substituted for those of hybrid Notch ligands. This analysis showed that only in DI-DLL1 and DI-JAG1 flies but not in DI-DLL4 and DI-JAG2 flies is the embryonic lethality rescued, the results being in agreement with the hybrid DI-DLL experiments on wing discs reported earlier in this work.

The authors conclude that their findings suggest that the activation mechanism of Notch during development differs between *Drosophila* (where both Neur and Mib1 are required for Notch-related developmental processes) and mammals and that this could possibly explain the apparently lesser relevance of mammalian NEUR proteins for developmental Notch signalling.

Evidence and clarity

The manuscript is quite laconic but clearly written. The evidence presented by the authors, given the heterologous and in vitro nature (i.e. using mammalian or hybrid Notch ligands and mammalian E3 ligases thereof in *Drosophila* and cell cultures) of the study is generally trustworthy but limited in the sense that it probably does not allow definitive conclusions to be drawn as to the differing nature of the action of the E3 ligases of Notch ligands in flies vs mammals in vivo.

Reproducibility

As will be mentioned a number of times, these reviewers would like to enquire as to the reasons for not providing a statistical analysis of variation in the fly wing disc-based experiments (where the readout was either rescue of Wg expression or induction of ectopic Wg expression). Also, while the constructs used in the study were inserted into the same genomic landing sites to achieve comparable levels of expression of the various proteins, these reviewers would like to see data on the levels of expression of NEUR1 and 2 as well as the hybrid Notch ligands.

****Major comments****

Comment on fly wing disc experiments:

The authors study both the capability of two different mammalian E3 ubiquitin ligases, Neuralized-like 1 and 2 (mouse Neur1 and human NEUR2) to activate four different Notch receptors (DLL1 and 2, JAG1 and 2) in flies and mammalian cell culture system. In flies, they first analyse the capability of *Drosophila* Neur (as a positive control) and Neur1 and NEUR2 to activate the various Notch ligands (based on wingless activation as a readout) in wild-type wings (where, Mind bomb 1, or Mib1 is the only E3 ligase for Notch ligands present) and Mib1 mutant wing discs (which lack any E3 ligands of Notch receptors). The authors then test four humanised, hybrid Notch ligands (all five N ligands bar Dll3 since the latter does not transactivate the Notch receptor) - where mammalian Notch ligands' intracellular domains, or ICDs, have been attached to fly DI (DI-DLL1, DI-DLL4, DI-JAG1, DI-JAG2) - for their capacity to mediate Mib1-dependent activation of Notch (with ectopic Wg expression in wing discs as its readout). They found that all 4 ligands can activate Notch in wild-type wings (where Mib1 is present), with DI-JAG2 being less effective than the other 3 hybrid ligands, implying that such hybrid, humanised ligands can be used in studying Notch pathway activation in *Drosophila* (thereby constituting a mixed/heterologous experimental system). The reviewers would like to get a comment as to the reason for the weaker activity of DI-JAG2 in this set-up?.

Also, the reviewers would like to get a comment as to why was not a Neur mutant set-up used, only Mib1 mutant discs?

The authors then found that only two of these hybrid ligands - DI-DLL1 and DI-JAG1 but not DI-DLL4 or DI-JAG2 - can be used to activate Notch in the above wing assay when Mib1 was mutant. This is consistent with the fact that the NxxN-based Neuralized Binding motif (NBM) is present in DLL1 and JAG1 only. Using the wing paradigm, the authors also show by either mutating the full NBM (NxxN) in DLL1 or changing the cryptic, "weak" NBM in DLL4 (containing NxxD sequence) into "full/strong" NxxN one that the NBM in the various Notch

ligands is required and sufficient for activation of the Notch pathway.

Overall, the fly experiments are convincing in showing differential activation of Notch ligands. However, no statistical analysis of the experimental variation in these studies - neither for the number of wing discs analysed per (hybrid) Notch ligand tested nor the extent of a given experimental manipulation's effect is included. We deem that if the images presented in Figures 2 and 3 are truly representative, this needs to be made explicitly clear.

Comment on fly embryonic Delta neurogenic phenotype's rescue experiments by replacing DL with the hybrid ligands:

The authors analysed the capacity of the ICDs of the mammalian ligands to rescue the DL phenotype in *Drosophila*, ie. their activation capability at the organismal level. This was achieved by generating knock-in alleles expressing the hybrid ligands in place of DL. The notion that only NBM-containing hybrid ligands was strengthened by this analysis since it showed that only NBM-containing hybrid ligands - DL-DLL1 and DL-JAG1 - but not DL-DLL4 nor DL-JAG2 rescued the DL neurogenic embryonic lethal phenotype. Since this experimental set-up relied on the endogenous *Drosophila* E3 ligases for activating the Notch ligands, the capacity of mammalian NEUR1 and 2 proteins to complement the capacity of the hybrid ligands to activate Notch to activate these ligands was not addressed. Please comment as to the reasons for this apparent omission and if such an analysis lies beyond the scope of current work, what would be the expected results of such experiment in the light of other experiments conducted in the course of this work?

Journal-agnostic peer review: evaluate the paper as it stands independently from potential journal fit.

Are the claims and the conclusions supported by the data or do they require additional experiments or analyses to support them?

Generally yes, but please see the above comments on the absence of statistical analysis of reproducibility/ variation (if any) in fly wing disc experiments.

****Reviewer's additional recommendations:****

To publish in a higher-ranking journal, the co-localisation analyses of Notch ligands and its various E3 ubiquitin ligases studied probably needs to be replaced by a more rigorous, ideally FRET-based approach.

Since previous studies have shown that the Notch ligands are (mostly) poly- or mono-ubiquitylated by the E3 ubiquitin ligases Mib and the NEUR proteins, ideally, this or its

follow-up study would benefit from analysis of the ubiquitylation status of the various hybrid Notch ligands.

Also, it would be useful to show the strength of interaction between the hybrid Notch ligands and NEUR1 and NEUR2 by using a co-immunoprecipitation based approach. Please request additional experiments only if they are essential for the conclusions. Alternatively, ask the authors to qualify their claims as preliminary or speculative, or to remove them altogether.

These reviewers do not strictly request any further experiments. However, since the mammalian NEUR2 could not be studied in cell cultures of U2OS cells due to its toxicity, we would like the authors to explain the choice of this cell line. Perhaps a cell line whose viability is not impaired by NEUR2 should be (or should have been) used?

If you have constructive further reaching suggestions that could significantly improve the study but would open new lines of investigations, please label them as "OPTIONAL".

As mentioned above, the NEUR2's capacity to activate the hybrid ligands in U2OS cells could not be addressed due to its toxicity. A more optimal cell line will have to be used in follow-up studies.

Also, these findings ultimately warrant in vivo studies using mice to definitively ascertain whether they also hold equally true there.

Are the suggested experiments realistic in terms of time and resources? It would help if you could add an estimated time investment for substantial experiments.

The suggested experiments are optional apart from statistical analysis of variation (if any) in the fly wing disc experiments. If there is no (apparently significant) variation in these data, this needs to be explicitly stated.

Are the data and the methods presented in such a way that they can be reproduced?

Generally yes, but see above about the lack of statistical data on the variation (if any).

Are the experiments adequately replicated and statistical analysis adequate?

Generally yes, but again, please see above about the lack of statistical data on the variation (if any).

****Minor comments****

Comment#1 (on the abstract and introduction):

In the Abstract, the authors state that there are four Notch ligands in mammals (lines 21 and 22):

"Thus, it is unclear how NEURL proteins regulate the four mammalian Notch ligands".

In the Introduction, they correctly state that there are five Notch ligands in mammals (lines 38 and 39):

„In mammals, there are five ligands, three from the Delta-like (Dll) family (Dll1, Dll3, Dll4), and two from the Jagged (Jag) family (Jag1 and Jag2)."

There are five Notch ligands in mammals (Dll1, Dll3, Dll4, Jag1, Jag2), and it is obvious that the authors are very well aware of this (they state in lines 146-147):

"We excluded the ICD of DLL3 since it is not a ligand capable of trans-activation of Notch" (the four ligands included were Dll1, Dll4, Jag1 and Jag2)."

Therefore, a clarification is required in the part of Abstract (i.e lines 21 and 22) - did the authors mean the four mammalian Notch ligands they actually studied (i.e Dll1, Dll4, Jag1, Jag2) or is there an oversight and the authors actually intended to write "the five Notch ligands in mammals"? In either case, a correction is required in this reviewer's opinion.

Specific experimental issues that are easily addressable.

NEUR2 could not be studied in mammalian cell cultures due to its toxicity in the U2OS cell line, the one used by the authors. The use of another cell line would not be probably overly time-consuming; however, if this experiment lies outside the scope of current work, we would like to hear the authors' comment on this matter.

Are prior studies referenced appropriately?

Generally yes, but four prior studies go unmentioned: the two 2001 mouse Neur1 knock-out studies reporting no Notch-like developmental phenotype (Ruan et al, PNAS; Vollrath et al, Mol Cell Biol), the 2002 study of mouse, rat and human NEUR1 expression, subcellular localisation (Timmusk et al, Mol Cell Neuroscience) and the 2009 cell culture-based study of NEUR2's interaction with DLL1 and DLL4 (Rullinkov et al, BBRC).

The non-requirement of NEUR1 and 2 proteins in mammalian developmental Notch signalling could partly be explained by the fact that NEUR1 is not highly expressed during mouse embryonic/foetal development - its expression becomes considerably more pronounced only postnatally (Timmusk et al, 2002).

Are the text and figures clear and accurate?

Yes. These reviewers find the cartoon-based explanations of the experimental set-up in each figure helpful for enhancing the manuscript's overall clarity.

Do you have suggestions that would help the authors improve the presentation of their data and conclusions?

Please see above about the lack of statistical data on the variation (if any) in fly wing disc experiments and referencing of the 4 papers that are currently excluded.

2. Significance:

Significance (Required)

Provide contextual information to readers (editors and researchers) about the novelty of the study, its value for the field and the communities that might be interested.

The following aspects are important:

General assessment: provide a summary of the strengths and limitations of the study. What are the strongest and most important aspects? What aspects of the study should be improved or could be developed?

This study uses the amenability of *Drosophila* to study the mammalian NEUR proteins' (NEUR1 and NEUR2) activity upon Notch ligands using hybrid Notch ligands containing mammalian ICDs (intracellular domains) fused to the extracellular domain of *Drosophila* Delta (Dl). It confirms and extends prior studies showing that Notch ligands can be (strongly) activated only by the E3 ubiquitin ligases containing the Neuralized Binding Motif (NBM).

However, since this study was based on using hybrid ligands containing mammalian ICDs of Notch ligands fused to the extracellular domain of *Drosophila* Delta (Dl), it is somewhat artificial. While NEUR1 was also studied in mammalian cell cultures (but not NEUR2 due to its toxicity), only an *in vivo* study using mice expressing with systematic changes to the Notch ligands' NBM will definitively reveal whether the conclusions reached by the authors hold true *in vivo* in a non-heterologous system.

Advance: compare the study to the closest related results in the literature or highlight results reported for the first time to your knowledge; does the study extend the knowledge in the field and in which way? Describe the nature of the advance and the resulting insights (for example: conceptual, technical, clinical, mechanistic, functional,...).

The study's advances are chiefly mechanistic and functional since they show more definitively that the reason underlying the differing activation of four mammalian Notch ligands by mammalian NEUR1 and NEUR2 is mostly based upon the presence or otherwise of a conserved Neuralized Binding Motif, NBM.

Audience: describe the type of audience ("specialised", "broad", "basic research", "translational/clinical", etc...) that will be interested or influenced by this research; how will this research be used by others; will it be of interest beyond the specific field?

The audience for this study is the research studying the Notch signalling pathway. Since dysregulation of this pathway is implicated in a number of devastating diseases, any improved understanding of its mechanistic underpinnings could in the long run lead to better therapeutic management of diseases with significant involvement of malfunctioning Notch signalling.

Please define your field of expertise with a few keywords to help the authors contextualize your point of view. Indicate if there are any parts of the paper that you do not have sufficient expertise to evaluate.

Molecular biology, molecular neuroscience, developmental biology, cell-cell signalling, Notch signalling. All parts of the manuscript fall within our expertise.

3. How much time do you estimate the authors will need to complete the suggested revisions:

Estimated time to Complete Revisions (Required)

(Decision Recommendation)

Between 1 and 3 months

No

Revision Plan

Manuscript number: RC-2024-02705

Corresponding author(s): David, Sprinzak

1. General Statements

*We are happy to submit a revision plan for our manuscript titled “Neuralized-like proteins differentially activate Notch ligands,” that sheds light on a new mechanism underlying the differential activity of Notch signaling in mammals. As with other major signaling pathways, Notch signaling involves multiple receptors and ligands capable of interacting with each other. However, it is unclear how specific interactions are regulated to produce different signals in different contexts. In the current manuscript, we used innovative ‘humanized flies’ and quantitative cell culture assays to unravel a novel mechanism in which the E3 ubiquitin ligases *Neurl1* and *Neurl2* selectively activate two Notch ligands (*Dll1* and *Jag1*), but not the other two active ligands (*Dll4* and *Jag2*). This differential activity is mediated by a Neuralized binding motif, *NxxN*, which is present in the two active ligands and absent in the non-active ones. This discovery is of general interest to the Notch and cell-cell signaling communities, as well as to the broad community interested in ubiquitin regulation.*

*The three reviewers provided an overall positive assessment of this manuscript, acknowledging both the significance and scientific rigor of the work. The comments raised by the reviewers fall into two main categories: (1) Statistical analysis of the *Drosophila* data - which we outline a plan below to address these concerns. (2) Confirmation/supportive experiments and clarifications - which are generally straightforward and we provide a plan for addressing them as well.*

Reviewer 3 also provided additional comments (not requirements) suggesting alternative experiments (see below) though he explicitly states these are not mandatory. We feel that these suggestions, while interesting, are not essential for substantiating the claims in the paper.

We are confident that the planned revisions will fully address the reviewers’ comments and suggestions to their satisfaction.

2. Description of the planned revisions

Please find below a point-by-point response to the reviewers’ comments. To make it clear we first provide a list of the planned experiments and analysis to address the major comments:

List of planned experiments/analysis

1. Quantitative/statistical analysis of *Drosophila* experiments. Reviewers 2 and 3 requested quantitative analysis of *Drosophila* wing images to demonstrate reproducibility across multiple samples. In response, we will include this analysis in the revised manuscript,

Revision Plan

analyzing n=5 discs per experiment and quantifying shape parameters. Statistical analysis will also be provided as requested.

2. Both reviewers 2 and 3 also suggested (but did not require) performing complementary co-IP experiments to further support the interactions between Neurl proteins and DLL1 and JAG1 but not DLL4 and JAG2. We plan to perform these experiments and include them in the revised manuscript.
3. Reviewer 3 proposed using FRET instead of our co-localization assays. While we believe our co-localization assays are functionally important, we acknowledge that they are not direct indications of molecular interaction. FRET, however, poses challenges in this experimental setup for several reasons. As an alternative, we propose using proximity ligation assay (PLA) to examine these interactions.
4. Reviewer 1 suggested investigating the competition between the two E3 ubiquitin ligase families, Mib1 and Neurls. While this broader question falls beyond the scope of the current work, we will conduct the control experiments recommended by the reviewer, where we express Neurl1 with Notch ligands in the presence of endogenous Mib1. This experiment may offer initial insights into what happens when both ligases are active.
5. The reviewers also requested additional discussion regarding ligand activity at endogenous levels. To strengthen the endogenous knockin data, we will include additional results analyzing a knock-in allele that encodes a DI-DLL4 hybrid with an introduced Neutralized binding motif. This experiment will clearly demonstrate that the rescue of the mutant DI phenotype by a DI-hybrid variant is dependent on the presence of the Neutralized binding motif.

Point-by-point response to the reviewers' comments is attached at the end of the document

3. Description of the revisions that have already been incorporated in the transferred manuscript

No revised manuscript is included with this plan.

4. Description of analyses that authors prefer not to carry out

Please include a point-by-point response explaining why some of the requested data or additional analyses might not be necessary or cannot be provided within the scope of a revision. This can be due to time or resource limitations or in case of disagreement about the necessity of such additional data given the scope of the study. Please leave empty if not applicable.

Revision Plan

None of the reviewers required additional experiments to support the core claims of our work. However, several suggestions were made, some of which are either unnecessary or beyond the scope of the current manuscript:

1. Reviewer 3 inquired why we did not use other cell lines to test *Neur12*, which proved toxic to the cell line we used (though the reviewer did not request to perform these experiments). We clarify that such experiments require a *Mib1* knock-out cell line. Generating a new *MIB1* knockout cell line and then screening for a suitable sender cell line is a time-intensive process that extends beyond the scope of the current work.
2. Reviewer 3 also asked why fly experiments were performed only in *Mib1* mutant background and not in a *Neur* mutant background. We address this by explaining that *Neur* expression occurs only at late developmental stages and is limited to a small subset of specific cells in the disc. Consequently, mutating *Neur* in our experiments is unnecessary.
3. Reviewer 3 further asked why we did not attempt to rescue endogenous *Neur* function in *Neur* mutant flies using mammalian *Neur11* and *Neur12*. We consider such experiments beyond the scope of the current manuscript, as it is highly unlikely that mammalian *NEURLs* proteins will fully compensate for the function of the *Drosophila* *Neur* mutant.

Point-by-point response to the reviewers comments

Reviewer #1 (Evidence, reproducibility and clarity (Required)):

This is an interesting manuscript from two groups of experts in Notch signaling biology with complementary expertise in *Drosophila* genetics (Klein) and in biophysical studies of the Notch pathway (Sprinzak). The paper provides a cutting-edge structure-function dissection of the E3 ubiquitin ligase *Neuralized* and its mammalian homologs, *Neur11a* and *Neur11b*. The work is particularly relevant since the functions of mammalian *Neur11a* and *Neur11b* have been questioned, and more subtle altogether than those of fly *Neuralized* (as summarized by the authors in Fig. 1C). This is in part due to the dominant effects of the E3 ubiquitin ligase *Mindbomb1* (*Mib1*) in Notch ligand-expressing cells from mammalian systems. The authors use careful structure-function work in fly development (mostly wing imaginal discs) and in mammalian cell culture systems, including a clever approach to study the function of mammalian *Neur11a* and *Neur11b* and mammalian/fly Notch ligand hybrids in *Drosophila* to draw new conclusions about the function of *Neur11a/b*, showing that they can function as activators of Notch signaling mediated by the Notch ligands *Dll1* and *Jag1*, and not by *Dll4* and *Jag2*, tracing these differential effects to the recognition of a short NXXN consensus sequence in the N-terminal region of the ligand's intracellular domain.

We thank the reviewer for highlighting the novelty of our findings and experimental approach.

Revision Plan

Specific questions:

-The current title of the manuscript is not very information-rich and would not allow a reader to gather key information about the findings without reading at least the abstract. Could this be improved? For example, by referring to differential activation of individual Notch ligands, or some other more direct description of the key findings?

We appreciate the reviewer's suggestion; however, we believe that the general nature of the title is appropriate in this case.

-The authors design most key experiments documenting agonistic effects of Neurl1a/1b in a Mib1-deficient background, both in flies and in cell culture systems. This is understandable experimentally to isolate Neurl1a/b's effects in these experimental systems. However, this leaves open questions as to the prevailing effects of Neurl1a/b in cells that also express Mib1 (which the authors comment on in the discussion based on past findings, including some suggesting that Neurl1a/1b can function as Notch inhibitors through a ligand ubiquitination mechanism that may differ from their activating function).

Do the authors actually have data that could shed light on this discussion? For example, have they performed cell coculture assays in which Neurl1a or Neurl1b is co-expressed with a Notch ligand, but in the presence of Mib1? This condition seems to be systematically omitted from all the coculture experiments that are presented. It would be interesting to evaluate the net effect of Neurl1a/Neurl1b expression in a Mib1-sufficient system as well.

We have systematically removed MIB1 in our experiments because it activates all ligands, making its removal necessary to show the differential activity of Neurls. The question regarding competition between Mib1 and Neurls, as highlighted by the reviewer, is indeed intriguing. However, systematically investigating this competition would require varying the relative levels of the two proteins in a controlled manner, which is beyond the scope of this manuscript.

That said, we will perform the competition experiments suggested by the reviewers (co-expressing ligands with both Neurl1 and Mib1) and test their activity as controls. While these experiments may provide some insight into the competition, they will not comprehensively address the entire topic.

-The paper suggests important predictions about mammalian functions of Neurl1a/1b, including the neurological effects that have been reported, in double-deficient mice, namely that there are cells that only express Neurl1a/1b and not Mib1 and do rely on Dll1 and Jag1 for signaling. Could the authors at least comment on this prediction? Are there any single cell atlases where candidate cells like that can be identified? Or would the authors predict that Neurl1a/1b could actually function as Notch agonist even in cells expressing Mib1? (see also previous comment)

This is an interesting suggestion. We will try to find if we can identify any specific expression patterns of E3 ubiquitin ligases across different tissues.

Revision Plan

-Some minor typos: line 305 should likely read "flies homozygous for (...)". Line 408, "for providing" repeated twice.

We thank the reviewer for pointing out this typo.

Reviewer #1 (Significance (Required)):

Thank you for the opportunity to review this lovely collaborative paper. As indicated in my comments to the authors, the findings provide novel structure-function information about an understudied aspect of Notch signaling and clarify conflicting past data about the mammalian homologs of fly Neuralized. The approach is elegant and multidisciplinary, notably in regards to the combination of cell co-culture systems and *Drosophila* as a platform to study mammalian Neuralized proteins and hybrid Notch ligand molecules. The findings will be interesting to the field and will generate discussion. I would suggest that some additional information would be a plus to substantiate predictions about mammalian functions of Neurl1a/b, and also to clarify its effects in the presence or absence of concomitant Mib1 expression.

We thank the reviewer for their positive evaluation of our work and for suggesting potential future direction regarding the concomitant expression of Mib1 and Neurls proteins.

Reviewer #2 (Evidence, reproducibility and clarity (Required)):

Summary

The manuscript describes an analysis of specificity of functional interactions between mammalian Neuralized proteins and different human ligands for Notch. To investigate this, the authors take the approach of constructing hybrid proteins that contain the intracellular domain of the human ligands and the extracellular domain of the *Drosophila* Delta or Serrate, and investigate their activity *in vivo*, in the *Drosophila* wing disc. The latter is a well-established model tissue for assessing Notch ligand activity. As a second assay they express mammalian neutralized constructs in human cells for luciferase-based Notch signal reporter assays. The experiments are well presented and described and make a strong case for the conclusions that both Neurl1 and 2 can activate Notch signalling by Dll1 and Jag1 but not Dll4 and Jag2. Use of different mutant intracellular domains is used to show the importance of the NXXN motif, which in *Drosophila* is required for Neuralized interaction with Delta and Serrate. The use of missense mutations and in particular the reactivation of the cryptic NXXD site in Dll4 by substitution to N is convincing for establishing the importance of the motif. There is also colocalization data to support the conclusion that there is likely to be NXXN-dependent complex formation between the ligand and Neuralized proteins. This latter conclusion would be made firmer if there were pull down data to support it, although to be fair it is most unlikely that another explanation, other than complex formation could account for the observation of both colocalization and ligand activation.

Revision Plan

We appreciate the reviewer's positive assessment of our manuscript and their support for the conclusions drawn from our experiments. We intend to conduct the suggested co-IP experiments with our cell culture assays to further supplement our current data.

Major comments

The main limitation of the work is that it is mostly based on overexpression of constructs to activate ectopic expression rather than gene editing endogenous genes. It would be helpful if the authors could comment on the limitations of the work in discussion.

Two points of data included in the work are important in mitigating this limitation. Firstly, the experiments in the wing disc and cell culture are taking place in a mindbomb mutant background and the activation is observed is therefore a rescue of activity that has been lost.

Secondly, and importantly, the final experiment makes use of a DI mutant Drosophila line which shows embryo lethality when homozygous, with the characteristic neurogenic phenotype. Rescue of lethality can be brought about by knock-in experiments which restore DI function and this is also true for the ligand hybrid constructs that introduce mammalian ligand intracellular domains only when they include the NXXN motif This indicates the importance of the motif in normal development-

Overall, the data presented in the paper is convincing as regards the conclusions made.

We thank the reviewer for their very positive evaluation and his constructive suggestions, which have helped to improve the manuscript. In line with these suggestions, we will include additional data analyzing the bristle SOP selection, a process dependent on Neur. Our Results show that homozygosity of the DlattP-DI-DLL1 allele, but not the DlattP-DI-DLL4 allele, leads to correct Notch mediated selection. This finding provides further evidence that Neur requires the NxxN motif in the ICD of a ligand to activate DSL ligands. Notably, we previously showed that this selection relies on the NxxN motif of DI (Troost et al., 2023).

We will further emphasize in the discussion the ability of DI-DLL4 hybrid ligands, containing a reconstructed NxxN motif, to rescue the neurogenic phenotype of DI mutants.

Minor points

In figure 1 the legend for D says that cryptic sites are substitutions of N for E or Q, but the figure and main text indicate that the substitutions are N to E or D.

We thank the reviewer for pointing this out. We will correct this mistake.

In the remain figures it would be helpful to include in the figure legends and indications of the numbers of wing discs, embryos for which the images shown are representative of.

We will quantify the experiments conducted in the wing imaginal discs of Drosophila by measuring the wing field size along the dorsal-ventral axis relative to the anterior-posterior axis. Statistical analysis will be performed to demonstrate statistical significance across n=5 experiments for each sample.

Revision Plan

In Fig 3 The activation of Notch, by neural1 and D1-Jag1 in B''' is stronger in the ventral side of the disc than the dorsal whereas, although activation of the same ligand by Neurl2 in C''' is weaker the majority of the ectopic wingless expression is on the dorsal compartment. Is there any reason for the switch in preference between the two neutralized proteins? Overgrowth of the wing disc seems to be similar on both sides and so am wondering if the picture is representative of the ectopic wingless distribution in this case.

As discussed above we will perform quantification and statistical analysis across multiple experiments to confirm that our images are truly representative.

Reviewer #2 (Significance (Required)):

Significance

Previous work on double genetic knockouts of the two mouse Neuralized genes cast doubt as to whether Neuralized proteins play a role in Notch signal activation in mammals, unlike in Drosophila. There is, however, some genetic indications that spatial memory requires both Notch and neutralized proteins and may represent a specialised function limited to the Neuralized interaction. There are likely to be more subtle contexts waiting to be uncovered. The work is therefore showing important proof of principle for establishing the functionality of the mammalian Neurl proteins and highlights new findings indicting specialisation of the different ligands for interactions with Notch components. Elucidation of such specialisations will help understand why the diversity of different homologues of Notch and ligand have evolved and are maintained in the vertebrate genome compared to the single Notch and two ligands in Drosophila. Since Notch and its misregulation are widely involved in development, health and disease and there is much interest in developing therapeutic interventions that alter Notch activity then the work is likely of broad interest.

We thank the reviewer for the very positive evaluation and his useful suggestions which were helpful in improving the manuscript.

Revision Plan

Reviewer #3 (Evidence, reproducibility and clarity (Required)):

****Summary****

Notch signalling is one of the major evolutionarily conserved signalling pathways involved in numerous developmental, physiological and pathological processes. Activation of the Notch receptor first requires ubiquitination of its ligands (collectively termed DSL), leading to a "pulling force" that, upon ligand-receptor engagement, exposes Notch to intramembrane proteolysis leading to the nuclear translocation of the receptor's intracellular domain and activation of target genes with its DNA-binding co-factors.

While both Neuralized (Neur1) and Mind bomb are the E3 ubiquitin ligases for Notch ligands required for Drosophila development, in mammals, the Neur homologues Neur1 (officially Neur1) and Neur2 (officially Neur1B) are dispensable for development since double Neur1/2 knock-out mice have no developmental phenotype (but both Neur homologues are involved in the memory-related functions of Notch pathway in adulthood). Rather, just one of the two mammalian Mind bomb homologues, Mib1, functions as the chief E3 ligase for Notch ligands during mammalian development as evidenced by its Notch-related knockout phenotype.

Therefore, it has not been fully established whether and how the NEUR proteins regulate the mammalian Notch ligands.

To clarify this issue, the authors assessed the capability of Drosophila Neur and mammalian NEUR1 and 2 proteins to activate the various hybrid Notch ligands (containing extracellularly Drosophila Delta and intracellularly the various Notch ligands' intracellular domains) in Drosophila wing discs and mammalian cell culture. The authors found that NEUR proteins only activate the Notch ligands containing a Neuralized binding motif, with the consensus sequence NxxN, that is present in DLL1 and JAG1, but not in DLL4 and JAG2.

The authors also analysed the intracellular domains of mammalian Notch ligands DLL1, DLL4, JAG1 and JAG2 in Drosophila by generating knock-in alleles where endogenous DI expression had been substituted for those of hybrid Notch ligands. This analysis showed that only in DI-DLL1 and DI-JAG1 flies but not in DI-DLL4 and DI-JAG2 flies is the embryonic lethality rescued, the results being in agreement with the hybrid DI-DLL experiments on wing discs reported earlier in this work.

The authors conclude that their findings suggest that the activation mechanism of Notch during development differs between Drosophila (where both Neur and Mib1 are required for Notch-related developmental processes) and mammals and that this could possibly explain the apparently lesser relevance of mammalian NEUR proteins for developmental Notch signalling.

Evidence and clarity

The manuscript is quite laconic but clearly written. The evidence presented by the authors,

Revision Plan

given the heterologous and in vitro nature (i.e using mammalian or hybrid Notch ligands and mammalian E3 ligases thereof in Drosophila and cell cultures) of the study is generally trustworthy but limited in the sense that it probably does not allow definitive conclusions to be drawn as to the differing nature of the action of the E3 ligases of Notch ligands in flies vs mammals in vivo.

We thank the reviewer for their positive evaluation of our work and their constructive criticism. We would like to clarify that we do not conclude that the activation mechanism differs between mammals and flies. Our findings demonstrate that the signalling mechanisms of fly Neur and mammalian Neurl's follow the same fundamental rules. Moreover, our study does not aim to provide a definitive answer to how signalling differs between species. Instead, we utilized the 'humanized fly' system to show that Neurl proteins specifically activate Dll1 and Jag1, but not Dll4 and Jag2, which lack a neuralized binding site.

Reproducibility

As will be mentioned a number of times, these reviewers would like to enquire as to the reasons for not providing a statistical analysis of variation in the fly wing disc-based experiments (where the readout was either rescue of Wg expression or induction of ectopic Wg expression).

We thank the reviewer for raising this important point. As outlined below, we will quantify the fly experiments and conduct statistical analysis across multiple experimental datasets to further substantiate our claims.

Also, while the constructs used in the study were inserted into the same genomic landing sites to achieve comparable levels of expression of the various proteins, these reviewers would like to see data on the levels of expression of NEUR1 and 2 as well as the hybrid Notch ligands.

Major comments

Comment on fly wing disc experiments:

The authors study both the capability of two different mammalian E3 ubiquitin ligases, Neuralized-like 1 and 2 (mouse Neur1 and human NEUR2) to activate four different Notch receptors (DLL1 and 2, JAG1 and 2) in flies and mammalian cell culture system. In flies, they first analyse the capability of Drosophila Neur (as a positive control) and Neur1 and NEUR2 to activate the various Notch ligands (based on wingless activation as a readout) in wild-type wings (where, Mind bomb 1, or Mib1 is the only E3 ligase for Notch ligands present) and Mib1 mutant wing discs (which lack any E3 ligands of Notch receptors). The authors then test four humanised, hybrid Notch ligands (all five N ligands bar Dll3 since the latter does not transactivate the Notch receptor) - where mammalian Notch ligands' intracellular domains, or ICDs, have been attached to fly DI (DI-Dll1, DI-Dll4, DI-JAG1, DI-JAG2) - for their capacity to

Revision Plan

mediate Mib1-dependent activation of Notch (with ectopic Wg expression in wing discs as its readout). They found that all 4 ligands can activate Notch in wild-type wings (where Mib1 is present), with DI-JAG2 being less effective than the other 3 hybrid ligands, implying that such hybrid, humanised ligands can be used in studying Notch pathway activation in Drosophila (thereby constituting a mixed/heterologous experimental system). The reviewers would like to get a comment as to the reason for the weaker activity of DI-JAG2 in this set-up?.

We do not have a definitive answer as to why the ICDs differ in their activity within Mib1-dependent signalling, since this question was not addressed in the scope of this work. However, our findings demonstrate that the hybrid ligands are functional in Drosophila and that their differential behavior in Neur-mediated signaling is not attributed to a trivial explanation, e. g. that the hybrid ligands generally display no activity. There are several potential explanations for these differences. One possibility is variations in position, arrangement, or number of targeted lysines among the ICDs. These lysines serve as substrates for ubiquitylation and determine the rate of endocytosis, which in turn impacts the signaling activity of the corresponding ligand/hybrid. Another plausible explanation is differences in affinity of the binding sites of Mib1, which would similarly result in variations in ubiquitylation and endocytosis rates. Regardless, we emphasize that resolving this question does not affect any of the conclusions of the manuscript.

Also, the reviewers would like to get a comment as to why was not a Neur mutant set-up used, only Mib1 mutant discs?

Neur is only expressed at a very late stage in wing development and is restricted to specific single cells (sensory organ precursors). Consequently, even if mutants were present, their impact would be limited to these cells. Moreover, the Neur promoter has a highly complex architecture, which makes it exceedingly difficult to manipulate for experiments involving this mutation. We will address these considerations in the revised manuscript.

The authors then found that only two of these hybrid ligands - DI-DLL1 and DI-JAG1 but not DI-DLL4 or DI-JAG2 - can be used to activate Notch in the above wing assay when Mib1 was mutant. This is consistent with the fact that the NxxN-based Neuralized Binding motif (NBM) is present in DLL1 and JAG1 only. Using the wing paradigm, the authors also show by either mutating the full NBM (NxxN) in DLL1 or changing the cryptic, "weak" NBM in DLL4 (containing NxxD sequence) into "full/strong" NxxN one that the NBM in the various Notch ligands is required and sufficient for activation of the Notch pathway.

Overall, the fly experiments are convincing in showing differential activation of Notch ligands. However, no statistical analysis of the experimental variation in these studies - neither for the number of wing discs analysed per (hybrid) Notch ligand tested nor the extent of a given experimental manipulation's effect is included. We deem that if the images presented in Figures 2 and 3 are truly representative, this needs to be made explicitly clear.

Revision Plan

We thank the reviewer for their positive evaluation of our work and for the constructive comments, which we will consider and include into the manuscript. While we have repeated all experiments with multiple flies, we acknowledge the critique regarding the absence of statistical analysis.

To address this, we will quantify the experiments conducted in the wing imaginal discs of *Drosophila*. We will do that by measuring the wing field size along the dorsal-ventral axis relative to the anterior-posterior axis. We will perform statistical analysis to assess the statistical significance between experiments, using data from n=5 experiments for each sample.

Comment on fly embryonic Delta neurogenic phenotype's rescue experiments by replacing DI with the hybrid ligands:

The authors analysed the capacity of the ICDs of the mammalian ligands to rescue the DI phenotype in *Drosophila*, ie. their activation capability at the organismal level. This was achieved by generating knock-in alleles expressing the hybrid ligands in place of DI. The notion that only NBM-containing hybrid ligands was strengthened by this analysis since it showed that only NBM-containing hybrid ligands - DI-DLL1 and DI-JAG1 - but not DI-DLL4 nor DI-JAG2 rescued the DI neurogenic embryonic lethal phenotype. Since this experimental set-up relied on the endogenous *Drosophila* E3 ligases for activating the Notch ligands, the capacity of mammalian NEUR1 and 2 proteins to complement the capacity of the hybrid ligands to activate Notch to activate these ligands was not addressed. Please comment as to the reasons for this apparent omission and if such an analysis lies beyond the scope of current work, what would be the expected results of such experiment in the light of other experiments conducted in the course of this work?

Testing whether mammalian *Neur1* and *Neur2* can replace *Drosophila* *Neur* in an endogenous setting is an intriguing question; however, it falls outside the focus of this study. Performing such an experiment would be highly challenging due to complex and not well understood architecture of *Neur* gene in *Drosophila*. Additionally, we believe it is highly unlikely that the mammalian NEURLs proteins would fully compensate for the loss of function in a *Drosophila* *Neur* mutant.

Journal-agnostic peer review: evaluate the paper as it stands independently from potential journal fit.

Are the claims and the conclusions supported by the data or do they require additional experiments or analyses to support them?

Generally yes, but please see the above comments on the absence of statistical analysis of reproducibility/ variation (if any) in fly wing disc experiments.

Reviewer's additional recommendations:

Revision Plan

To publish in a higher-ranking journal, the co-localisation analyses of Notch ligands and its various E3 ubiquitin ligases studied probably needs to be replaced by a more rigorous, ideally FRET-based approach.

We thank the reviewer for the comment. The co-localization assay is quite a robust and functional approach, as it provides clear evidence that endocytosis into a different compartment has occurred with functional ligands, as opposed to non-functional ligands. This serves as a quantitative and rigorous indicator for functional differences between these ligand types.

Nevertheless, we acknowledge that co-localization is not a direct measure of molecular interactions between Neur1 and Notch ligands. To address this, as suggested by the reviewer, we will perform co-IP to show the differential interaction between Neur1 and specific Notch ligands. Additionally, we will attempt a proximity ligation assay (PLA), which we consider to be a more direct and suitable method for detecting interactions between NEURLs and Notch ligands in this context, compared to FRET.

Since previous studies have shown that the Notch ligands are (mostly) poly- or mono-ubiquitylated by the E3 ubiquitin ligases Mib and the NEUR proteins, ideally, this or its follow-up study would benefit from analysis of the ubiquitylation status of the various hybrid Notch ligands.

We thank the reviewer for the suggestion. The ubiquitylation pattern by Neur1 is beyond the topic of the current manuscript.

Also, it would be useful to show the strength of interaction between the hybrid Notch ligands and NEUR1 and NEUR2 by using a co-immunoprecipitation based approach.

As suggested by the reviewer, we plan to perform co-IP and/or PLA to show the differential binding of NEURL1 to the different ligands. However, due to the observed toxicity of NEURL2 in our cells, it has been excluded from our assays.

Please request additional experiments only if they are essential for the conclusions. Alternatively, ask the authors to qualify their claims as preliminary or speculative, or to remove them altogether.

These reviewers do not strictly request any further experiments. However, since the mammalian NEUR2 could not be studied in cell cultures of U2OS cells due to its toxicity, we would like the authors to explain the choice of this cell line. Perhaps a cell line whose viability is not impaired by NEUR2 should be (or should have been) used?

Revision Plan

The decision not to use other cell lines was based on several strict experimental requirements. The most stringent requirement was the need to generate a MIB1 knockout cell line, as MIB1 strongly activates all ligands. The availability of having MIB1 KO U2OS cells enabled these experiments.

If you have constructive further reaching suggestions that could significantly improve the study but would open new lines of investigations, please label them as "OPTIONAL".

As mentioned above, the NEUR2's capacity to activate the hybrid ligands in U2OS cells could not be addressed to due to its toxicity. A more optimal cell line will have to be used in follow-up studies.

Also, these findings ultimately warrant in vivo studies using mice to definitively ascertain whether they also hold equally true there.

Are the suggested experiments realistic in terms of time and resources? It would help if you could add an estimated time investment for substantial experiments.

The suggested experiments are optional apart from statistical analysis of variation (if any) in the fly wing disc experiments. If there is no (apparently significant) variation in these data, this needs to explicitly stated.

We thank the reviewer for their thoughtful assessment. We will conduct the requested statistical analysis and perform some of the suggested supporting experiments.

Are the data and the methods presented in such a way that they can be reproduced?

Generally yes, but see above about the lack of statistical data on the variation (if any).

Are the experiments adequately replicated and statistical analysis adequate?

Generally yes, but again, please see above about the lack of statistical data on the variation (if any).

****Minor comments****

Comment#1 (on the abstract and introduction):

In the Abstract, the authors state that there are four Notch ligands in mammals (lines 21 and 22):

"Thus, it is unclear how NEURL proteins regulate the four mammalian Notch ligands".

In the Introduction, they correctly state that there are five Notch ligands in mammals (lines 38

Revision Plan

and 39):

„In mammals, there are five ligands, three from the Delta-like (Dll) family (Dll1, Dll3, Dll4), and two from the Jagged (Jag) family (Jag1 and Jag2)."

There are five Notch ligands in mammals (Dll1, Dll3, Dll4, Jag1, Jag2), and it is obvious that the authors are very well aware of this (they state in lines 146-147):

"We excluded the ICD of DLL3 since it is not a ligand capable of trans-activation of Notch" (the four ligands included were Dll1, Dll4, Jag1 and Jag2)."

Therefore, a clarification is required in the part of Abstract (i.e lines 21 and 22) - did the authors mean the four mammalian Notch ligands they actually studied (i.e Dll1, Dll4, Jag1, Jag2) or is there an oversight and the authors actually intended to write "the five Notch ligands in mammals".? In either case, a correction is required in this reviewer's opinion.

We are fully aware of this point, and will address it by providing clarification in the abstract as suggested.

Specific experimental issues that are easily addressable.

NEUR2 could not be studied in mammalian cell cultures due to its toxicity in the U2OS cell line, the one used by the authors. The use of another cell line would not be probably overly time-consuming; however, if this experiment lies outside the scope of current work, we would like to hear the authors' comment on this matter.

This is addressed above.

Are prior studies referenced appropriately?

Generally yes, but four prior studies go unmentioned: the two 2001 mouse Neur1 knock-out studies reporting no Notch-like developmental phenotype (Ruan et al, PNAS; Vollrath et al, Mol Cell Biol), the 2002 study of mouse, rat and human NEUR1 expression, subcellular localisation (Timmusk et al, Mol Cell Neuroscience) and the 2009 cell culture-based study of NEUR2's interaction with DLL1 and DLL4 (Rullinkov et al, BBRC).

The non-requirement of NEUR1 and 2 proteins in mammalian developmental Notch signalling could partly be explained by the fact that NEUR1 is not highly expressed during mouse embryonic/foetal development - its expression becomes considerably more pronounced only postnatally (Timmusk et al, 2002).

We will incorporate these references into the introduction and discuss the low expression of Neurls during development as a possible reason for the non-requirement in this context.

Are the text and figures clear and accurate?

Yes. These reviewers find the cartoon-based explanations of the experimental set-up in each

Revision Plan

figure helpful for enhancing the manuscript's overall clarity.

We thank the reviewers for the positive feedback!

Do you have suggestions that would help the authors improve the presentation of their data and conclusions?

Please see above about the lack of statistical data on the variation (if any) in fly wing dic experiments and referencing of the 4 papers that are currently excluded.

These will be corrected in the revised version.

Reviewer #3 (Significance (Required)):

2. Significance

Provide contextual information to readers (editors and researchers) about the novelty of the study, its value for the field and the communities that might be interested.

The following aspects are important:

General assessment: provide a summary of the strengths and limitations of the study. What are the strongest and most important aspects? What aspects of the study should be improved or could be developed?

This study uses the amenability of *Drosophila* to study the mammalian NEUR proteins' (NEUR1 and NEUR2) activity upon Notch ligands using hybrid Notch ligands containing mammalian ICDs (intracellular domains) fused to the extracellular domain of *Drosophila* Delta (DI). It confirms and extends prior studies showing that Notch ligands can be (strongly) activated only by the E3 ubiquitin ligases containing the Neuralized Binding Motif (NBM).

We respectfully disagree with the reviewer's assessment on this point. Our study is the first to demonstrate that Neurl proteins differentially activate Dll1 and Jag1, but not Dll4 and Jag2. This findings is further supported by the significance comments of the other reviewers.

However, since this study was based on using hybrid ligands containing mammalian ICDs of Notch ligands fused to the extracellular domain of *Drosophila* Delta (DI), it is somewhat artificial. While NEUR1 was also studied in mammalian cell cultures (but not NEUR2 due to its toxicity), only an in vivo study using mice expressing with systematic changes to the Notch ligands' NBM will definitively reveal whether the conclusions reached by the authors hold true in vivo in a non-heterologous system.

Revision Plan

We firmly believe that our combined 'humanized fly' model and quantitative cell culture assay represents an innovative and rigorous approach for testing humanized proteins in in-vivo settings, without the need for extensive mouse genetics. The conclusions of our experiments should not be dismissed solely on the grounds of "not being performed in mice," as this would undermine much of current scientific research.

Advance: compare the study to the closest related results in the literature or highlight results reported for the first time to your knowledge; does the study extend the knowledge in the field and in which way? Describe the nature of the advance and the resulting insights (for example: conceptual, technical, clinical, mechanistic, functional,...).

The study's advances are chiefly mechanistic and functional since they show more definitively that the reason underlying the differing activation of four mammalian Notch ligands by mammalian NEUR1 and NEUR2 is mostly based upon the presence or otherwise of a conserved Neuralized Binding Motif, NBM.

Audience: describe the type of audience ("specialised", "broad", "basic research", "translational/clinical", etc...) that will be interested or influenced by this research; how will this research be used by others; will it be of interest beyond the specific field?

The audience for this study is the research studying the Notch signalling pathway. Since dysregulation of this pathway is implicated in a number of devastating diseases, any improved understanding of its mechanistic underpinnings could in the long run lead to better therapeutic management of diseases with significant involvement of malfunctioning Notch signalling.

Please define your field of expertise with a few keywords to help the authors contextualize your point of view. Indicate if there are any parts of the paper that you do not have sufficient expertise to evaluate.

Molecular biology, molecular neuroscience, developmental biology, cell-cell signalling, Notch signaling. All parts of the manuscript fall within our expertise.

Dear Prof. Sprinzak,

Thank you for transferring your manuscript to EMBO Reports, which was previously reviewed at Review Commons.

Referees express interest in your study investigating the function of Neuralized-like proteins in regulation of Notch ligand activation. However, they also raise concerns that need to be addressed to consider publication in EMBO Reports.

Having looked at all documents, we would like to invite you to submit a revised manuscript as in your revision plan. Please revise your manuscript with the understanding that the referee concerns (as in their reports) must be fully addressed and their suggestions taken on board. Please address all referee concerns in a complete point-by-point response. Acceptance of the manuscript will depend on a positive outcome of a second round of review. It is EMBO reports policy to allow a single round of major experimental revision only and acceptance or rejection of the manuscript will therefore depend on the completeness of your responses included in the next, final version of the manuscript.

We realize that it is difficult to revise to a specific deadline. In the interest of protecting the conceptual advance provided by the work, we recommend a revision within 3 months. Please discuss the revision progress ahead of this time with me if you require more time to complete the revisions, or if you have questions or comments regarding the revision (also by video chat).

1. A data availability section providing access to data deposited in public databases is missing (where applicable).
2. Your manuscript contains statistics and error bars based on $n=2$. Please use scatter plots in these cases.

You can submit the revision either as a Scientific Report or as a Research Article. For Scientific Reports, the revised manuscript can contain up to 5 main figures and 5 Expanded View figures, and it should not exceed 27000 characters. If the revision leads to a manuscript with more than 5 main figures it will be published as a Research Article. In this case the Results and Discussion section should be separate. If a Scientific Report is submitted, these sections have to be combined. This will help to shorten the manuscript text by eliminating some redundancy that is inevitable when discussing the same experiments twice. In either case, all materials and methods should be included in the main manuscript file.

4) a .docx formatted letter INCLUDING the reviewers' reports and your detailed point-by-point responses to their comments. As part of the EMBO publication's Transparent Editorial Process, EMBO reports publishes online a Review Process File (RPF) to accompany accepted manuscripts. This File will be published in conjunction with your paper and will include the referee reports, your point-by-point response and all pertinent correspondence relating to the manuscript.

<https://www.embopress.org/page/journal/14693178/authorguide#transparentprocess>

5) a complete author checklist, which you can download from our author guidelines <https://www.embopress.org/page/journal/14693178/authorguide>. Please insert information in the checklist that is also reflected in the manuscript. The completed author checklist will also be part of the RPF.

6) Please note that all corresponding authors are required to supply an ORCID ID for their name upon submission of a revised manuscript (). Please find instructions on how to link your ORCID ID to your account in our manuscript tracking system in our Author guidelines

Additional information on source data and instruction on how to label the files are available:
<https://www.embopress.org/page/journal/14693178/authorguide#sourcedata>

9) Our journal encourages inclusion of *data citations in the reference list* to directly cite datasets that were re-used and obtained from public databases. Data citations in the article text are distinct from normal bibliographical citations and should directly link to the database records from which the data can be accessed. In the main text, data citations are formatted as follows: "Data ref: Smith et al, 2001" or "Data ref: NCBI Sequence Read Archive PRJNA342805, 2017". In the Reference list, data citations must be labeled with "[DATASET]". A data reference must provide the database name, accession number/identifiers and a resolvable link to the landing page from which the data can be accessed at the end of the reference. Further instructions are available at <http://www.embopress.org/page/journal/14693178/authorguide#referencesformat>

12) Please also note our reference format:
<http://www.embopress.org/page/journal/14693178/authorguide#referencesformat>

13) All Materials and Methods need to be described in the main text using our 'Structured Methods' format, which is required for

all research articles. According to this format, the Methods section includes a Reagents and Tools Table (listing key reagents, experimental models, software and relevant equipment and including their sources and relevant identifiers) followed by a Methods and Protocols section describing the methods using a step-by-step protocol format. The aim is to facilitate adoption of the methodologies across labs. More information on how to adhere to this format as well as a downloadable template (.docx) for the Reagents and Tools Table can be found in our author guidelines:
<https://www.embopress.org/page/journal/14693178/authorguide#structuredmethods>.

An example of a Method paper with Structured Methods can be found here:
<https://www.embopress.org/doi/10.15252/msb.20178071>.

I look forward to seeing a revised version of your manuscript when it is ready. Please let me know if you have questions or comments regarding the revision.

Kind regards,

Deniz Senyilmaz Tiebe

Deniz Senyilmaz Tiebe, PhD
Senior Scientific Editor
EMBO Reports

Response To Reviewers:

We thank the reviewers for their thorough review of our manuscript and for their helpful comments that have certainly contributed significantly to the clarity of the manuscript.

Reviewer #1 (Evidence, reproducibility and clarity (Required)):

This is an interesting manuscript from two groups of experts in Notch signaling biology with complementary expertise in *Drosophila* genetics (Klein) and in biophysical studies of the Notch pathway (Sprinzak). The paper provides a cutting-edge structure-function dissection of the E3 ubiquitin ligase Neuralized and its mammalian homologs, Neurl1a and Neurl1b. The work is particularly relevant since the functions of mammalian Neurl1a and Neurl1b have been questioned, and more subtle altogether than those of fly Neuralized (as summarized by the authors in Fig. 1C). This is in part due to the dominant effects of the E3 ubiquitin ligase Mindbomb1 (Mib1) in Notch ligand-expressing cells from mammalian systems. The authors use careful structure-function work in fly development (mostly wing imaginal discs) and in mammalian cell culture systems, including a clever approach to study the function of mammalian Neurl1a and Neurl1b and mammalian/fly Notch ligand hybrids in *Drosophila* to draw new conclusions about the function of Neurl1a/b, showing that they can function as activators of Notch signaling mediated by the Notch ligands Dll1 and Jag1, and not by Dll4 and Jag2, tracing these differential effects to the recognition of a short NXXN consensus sequence in the N-terminal region of the ligand's intracellular domain.

We thank the reviewer for highlighting the novelty of our findings and experimental approach. We are also thankful for the reviewer for pointing out the annotation of NEURL proteins we used was not the formal one currently used in UNIPROT. We have changed our annotation to NEURL1 and NEURL1B which are the ones currently defined by UNIPROT throughout the manuscript.

Specific questions:

-The current title of the manuscript is not very information-rich and would not allow a reader to gather key information about the findings without reading at least the abstract. Could this be improved? For example, by referring to differential activation of individual Notch ligands, or some other more direct description of the key findings?

We appreciate the reviewer's suggestion; however, we believe that the general nature of the title is appropriate in this case.

-The authors design most key experiments documenting agonistic effects of Neurl1a/1b in a Mib1-deficient background, both in flies and in cell culture systems. This is understandable experimentally to isolate Neurl1a/b's effects in these experimental systems. However, this leaves open questions as to the prevailing effects of Neurl1a/b in cells that also express Mib1

(which the authors comment on in the discussion based on past findings, including some suggesting that Neurl1a/1b can function as Notch inhibitors through a ligand ubiquitination mechanism that may differ from their activating function).

Do the authors actually have data that could shed light on this discussion? For example, have they performed cell coculture assays in which Neurl1a or Neurl1b is co-expressed with a Notch ligand, but in the presence of Mib1? This condition seems to be systematically omitted from all the coculture experiments that are presented. It would be interesting to evaluate the net effect of Neurl1a/Neurl1b expression in a Mib1-sufficient system as well.

We have systematically removed MIB1 in our experiments because it activates all ligands, making its removal necessary to show the differential activity of Neurls. The question regarding competition between Mib1 and Neurls, as highlighted by the reviewer, is indeed intriguing. However, we feel that this question is beyond the scope of this manuscript as it requires a full set of experiments, where the relative levels of MIB1 and NEURLs is varied systematically.

Nonetheless, we did perform one set of competition experiments suggested by the reviewers (co-expressing ligands with both NEURL1 and endogenous MIB1) and tested their activity as controls (Fig. R1 below). We found that NEURL1 enhanced DLL1 signaling, suggesting cooperative rather than competitive interactions with MIB1. Co-expression with DLL4, JAG1, or JAG2 had no significant effect, possibly due to ligand-specific differences or pathway saturation. We note that while these results indicate potentially interesting and complex interactions between NEURL1 and MIB1 (That add up to other published results showing competition between NEURL1 and MIB1 in cell culture experiments), they can be interpreted in different ways and may depend on the relative expression of the two proteins within the cells. Proper understanding of the competitive/cooperative dynamics between MIB1 and NEURL1 will require carefully controlled titration experiments, and represents an intriguing area for further comprehensive investigation in the future. We therefore have chosen not to add this data to the current manuscript.

Figure for referee with unpublished data and its description has been removed upon request by the authors.

-The paper suggests important predictions about mammalian functions of Neurl1a/1b, including the neurological effects that have been reported, in double-deficient mice, namely that there are cells that only express Neurl1a/1b and not Mib1 and do rely on Dll1 and Jag1 for signaling. Could the authors at least comment on this prediction? Are there any single cell atlases where candidate cells like that can be identified? Or would the authors predict that Neurl1a/1b could actually function as Notch agonist even in cells expressing Mib1? (see also previous comment)

We thank the reviewer for this suggestion. Based on the reviewer's suggestion we have looked to determine whether NEURL proteins and MIB1 are differentially expressed in different human tissues and cells and whether there is a correlation with Notch ligand expression.

We analyzed expression data for MIB1, NEURL1, and NEURL1B across 54 human tissues from the Genotype-Tissue Expression (GTEx) Project database⁴¹ and single-cell RNAseq data from the Allen Brain Cell (ABC) Atlas⁴² (Appendix Fig. S7). We found that NEURL1 is highly expressed in specific tissues, particularly in the brain and skeletal muscle (Appendix Fig. S7A). In contrast, Mib1 and NEURL1B are more widely expressed across many tissues but at lower levels in the same tissues where NEURL1 is highly expressed (Appendix Fig. S7A, highlighted area). To take a closer look at the expression patterns within individual human neurons, we analyzed publicly available single-cell RNAseq data from the Allen Brain Atlas, which confirmed

similar trends. Uniform Manifold Approximation and Projection (UMAP) analysis identified 36 clusters corresponding to different neurons with different expression patterns (referred to as cell types, Appendix Fig. S7B). Most (but not all) of these cell types express high levels of NEURL1 and low levels of both MIB1 and NEURL1B (Appendix Fig. S7C-F) consistent with the tissue-level expression patterns observed in (Appendix Fig. S7A). Quantitative analysis of NEURL1, MIB1, NEURL1B, and the various Notch ligands in these cells revealed that DLL1 and JAG2 are expressed in most cell types, while JAG1 is found in some, and DLL4 is almost not expressed across all cell types. Overall, this analysis highlights differential expression patterns of NEURL proteins and MIB1 in tissues, with NEURL1 being more highly expressed in the brain, alongside lower expression levels of MIB1 and NEURL1B. Notably, high levels of NEURL1 expression are often correlated with DLL1 expression in many cell types, and in some instances, with JAG1 expression. These ligands can be activated by NEURL1. Interestingly, these same cell types also express JAG2, which cannot be activated by NEURL1, suggesting that other activation mechanisms might be at play.

We also add a short part in the discussion about this.

-Some minor typos: line 305 should likely read "flies homozygous for (...)". Line 408, "for providing" repeated twice.

We thank the reviewer for pointing out this typo, it was corrected.

Reviewer #1 (Significance (Required)):

Thank you for the opportunity to review this lovely collaborative paper. As indicated in my comments to the authors, the findings provide novel structure-function information about an understudied aspect of Notch signaling and clarify conflicting past data about the mammalian homologs of fly Neuralized. The approach is elegant and multidisciplinary, notably in regards to the combination of cell co-culture systems and *Drosophila* as a platform to study mammalian Neuralized proteins and hybrid Notch ligand molecules. The findings will be interesting to the field and will generate discussion. I would suggest that some additional information would be a plus to substantiate predictions about mammalian functions of Neurl1a/b, and also to clarify its effects in the presence or absence of concomitant Mib1 expression.

We thank the reviewer for their positive evaluation of our work and for suggesting potential future direction regarding the concomitant expression of Mib1 and Neurls proteins.

Reviewer #2 (Evidence, reproducibility and clarity (Required)):

Summary

The manuscript describes an analysis of specificity of functional interactions between mammalian Neuralized proteins and different human ligands for Notch. To investigate this, the authors take the approach of constructing hybrid proteins that contain the intracellular domain of the human ligands and the extracellular domain of the *Drosophila* Delta or Serrate, and investigate their activity *in vivo*, in the *Drosophila* wing disc. The latter is a well-established model tissue for assessing Notch ligand activity. As a second assay they express mammalian neutralized constructs in human cells for luciferase-based Notch signal reporter assays. The experiments are well presented and described and make a strong case for the conclusions that both *Neur1* and *2* can activate Notch signalling by *Dll1* and *Jag1* but not *Dll4* and *Jag2*. Use of different mutant intracellular domains is used to show the importance of the NXXN motif, which in *Drosophila* is required for Neuralized interaction with Delta and Serrate. The use of missense mutations and in particular the reactivation of the cryptic NXXD site in *Dll4* by substitution to N is convincing for establishing the importance of the motif. There is also colocalization data to support the conclusion that there is likely to be NXXN-dependent complex formation between the ligand and Neuralized proteins. This latter conclusion would be made firmer if there were pull down data to support it, although to be fair it is most unlikely that another explanation, other than complex formation could account for the observation of both colocalization and ligand activation.

Following the recommendation, we performed additional co-immunoprecipitation (co-IP) assays in our cell culture system to complement the existing data and added the data to the manuscript (New section, and Appendix Fig. 6S). Surprisingly, all tested ligand variants co-immunoprecipitated with NEURL1, and neither removal of the NxxN motif from *DLL1* and *JAG1* nor its introducing into *DLL4* significantly affected the binding interaction. These results suggest that, unlike the situation in *Drosophila* where mutation of the NBM in *DI* completely abolishes binding to *Neur* (ref 9), mammalian NEURL1 binds Notch ligands through a more complex mechanism. Thus, although the NxxN motif is essential for ligand activation and proper localization in mammalian cells, additional interaction sites or adaptor proteins are likely involved in stabilizing the ligand-NEURL1 complex.

Major comments

The main limitation of the work is that it is mostly based on overexpression of constructs to activate ectopic expression rather than gene editing endogenous genes. It would be helpful if the authors could comment on the limitations of the work in discussion.

Two points of data included in the work are important in mitigating this limitation. Firstly, the experiments in the wing disc and cell culture are taking place in a *mindbomb* mutant background and the activation is observed is therefore a rescue of activity that has been lost.

Secondly, and importantly, the final experiment makes use of a *DI* mutant *Drosophila* line which shows embryo lethality when homozygous, with the characteristic neurogenic phenotype. Rescue of lethality can be brought about by knock-in experiments which restore *DI* function and this is also true for the ligand hybrid constructs that introduce mammalian ligand intracellular domains only when they include the NXXN motif This indicates the importance of the motif in

normal development-

Overall, the data presented in the paper is convincing as regards the conclusions made.

We thank the reviewer for their very positive evaluation and their constructive suggestions, which have helped to improve the manuscript. We have added in the discussion the points raised by the reviewer that mitigate the limitations of the overexpression experiments.

In line with these suggestions, we have highlighted in our knock-in DI rescue experiments (Fig.6) that not only did the *DlattP-DLL1/JAG1* overcome embryonic lethality, another Neur-dependent process, namely bristle development, took place without any gross defects.

Minor points

In figure 1 the legend for D says that cryptic sites are substitutions of N for E or Q, but the figure and main text indicate that the substitutions are N to E or D.

We thank the reviewer for pointing this out. We corrected this mistake.

In the remain figures it would be helpful to include in the figure legends and indications of the numbers of wing discs, embryos for which the images shown are representative of.

We thank the reviewer for this comment. In the revised manuscript we have added quantification of the experiments conducted in the wing imaginal discs of *Drosophila* by measuring the over-proliferation of the wing disc along the dorsal-ventral axis relative to the anterior-posterior axis and the length of the Wg expression lines. Statistical analysis was performed to demonstrate statistical significance across n=5 experiments for each sample. They are now included in the updated manuscript in Fig. 2. G,H, Appendix Fig. 2SC, Fig. 3D, Appendix Fig. 3SA, Fig. 5D.

In Fg 3 The activation of Notch, by neural1 and DI-Jag1 in B''' is stronger in the ventral side of the disc than the dorsal whereas, although activation of the same ligand by Neural2 in C''' is weaker the majority of the ectopic wingless expression is on the dorsal compartment. Is there any reason for the switch in preference between the two neutralized proteins? Overgrowth of the wing disc seems to be similar on both sides and so am wondering if the picture is representative of the ectopic wingless distribution in this case.

As discussed above we added quantification and statistical analysis across multiple experiments and confirmed that our images are truly representative. They are included in the updated manuscript.

Reviewer #2 (Significance (Required)):

Significance

Previous work on double genetic knockouts of the two mouse Neuralized genes cast doubt as

to whether Neuralized proteins play a role in Notch signal activation in mammals, unlike in *Drosophila*. There is, however, some genetic indications that spatial memory requires both Notch and neuralized proteins and may represent a specialised function limited to the Neuralized interaction. There are likely to be more subtle contexts waiting to be uncovered. The work is therefore showing important proof of principle for establishing the functionality of the mammalian Neural proteins and highlights new findings indicating specialisation of the different ligands for interactions with Notch components. Elucidation of such specialisations will help understand why the diversity of different homologues of Notch and ligand have evolved and are maintained in the vertebrate genome compared to the single Notch and two ligands in *Drosophila*. Since Notch and its misregulation are widely involved in development, health and disease and there is much interest in developing therapeutic interventions that alter Notch activity then the work is likely of broad interest.

We thank the reviewer for the very positive evaluation and his useful suggestions which were helpful in improving the manuscript.

Reviewer #3 (Evidence, reproducibility and clarity (Required)):

****Summary****

Notch signalling is one of the major evolutionarily conserved signalling pathways involved in numerous developmental, physiological and pathological processes. Activation of the Notch receptor first requires ubiquitination of its ligands (collectively termed DSL), leading to a 'pulling force' that, upon ligand-receptor engagement, exposes Notch to intramembrane proteolysis leading to the nuclear translocation of the receptor's intracellular domain and activation of target genes with its DNA-binding co-factors.

While both Neuralized (Neur1) and Mind bomb are the E3 ubiquitin ligases for Notch ligands required for Drosophila development, in mammals, the Neur homologues Neur1 (officially Neur11) and Neur2 (officially Neur11B) are dispensable for development since double Neur1/2 knock-out mice have no developmental phenotype (but both Neur homologues are involved in the memory-related functions of Notch pathway in adulthood). Rather, just one of the two mammalian Mind bomb homologues, Mib1, functions as the chief E3 ligase for Notch ligands during mammalian development as evidenced by its Notch-related knockout phenotype.

Therefore, it has not been fully established whether and how the NEUR proteins regulate the mammalian Notch ligands.

To clarify this issue, the authors assessed the capability of Drosophila Neur and mammalian NEUR1 and 2 proteins to activate the various hybrid Notch ligands (containing extracellularly Drosophila Delta and intracellularly the various Notch ligands' intracellular domains) in Drosophila wing discs and mammalian cell culture. The authors found that NEUR proteins only activate the Notch ligands containing a Neuralized binding motif, with the consensus sequence NxxN, that is present in DLL1 and JAG1, but not in DLL4 and JAG2.

The authors also analyse the intracellular domains of mammalian Notch ligands DLL1, DLL4, JAG1 and JAG2 in Drosophila by generating knock-in alleles where endogenous DI expression had been substituted for those of hybrid Notch ligands. This analysis showed that only in DI-DLL1 and DI-JAG1 flies but not in DI-DLL4 and DI-JAG2 flies is the embryonic lethality rescued, the results being in agreement with the hybrid DI-DLL experiments on wing discs reported earlier in this work.

The authors conclude that their findings suggest that the activation mechanism of Notch during development differs between Drosophila (where both Neur and Mib1 are required for Notch-related developmental processes) and mammals and that this could possibly explain the apparently lesser relevance of mammalian NEUR proteins for developmental Notch signalling.

We thank the reviewer for highlighting the novelty of our findings and experimental approach. We are also thankful for the reviewer for pointing out the annotation of NEURL proteins we used was not the official one currently used in UNIPROT. We have changed our annotation to

NEURL1 and NEURL1B which are the ones currently defined by UNIPROT throughout the manuscript.

Evidence and clarity

The manuscript is quite laconic but clearly written. The evidence presented by the authors, given the heterologous and in vitro nature (i.e using mammalian or hybrid Notch ligands and mammalian E3 ligases thereof in Drosophila and cell cultures) of the study is generally trustworthy but limited in the sense that it probably does not allow definitive conclusions to be drawn as to the differing nature of the action of the E3 ligases of Notch ligands in flies vs mammals in vivo.

We thank the reviewer for their positive evaluation of our work and their constructive criticism. We would like to clarify that we do not conclude that the activation mechanism differs between mammals and flies. Our findings demonstrate that the signalling mechanisms of fly Neur and mammalian Neurl's follow the same fundamental rules. Moreover, our study does not aim to provide a definitive answer to how signalling differs between species. Instead, we utilized the 'humanized fly' system to show that Neurl proteins specifically activate DLL1 and JAG1, but not DLL4 and JAG2, which lack a neuralized binding motif.

Reproducibility

As will be mentioned a number of times, these reviewers would like to enquire as to the reasons for not providing a statistical analysis of variation in the fly wing disc-based experiments (where the readout was either rescue of Wg expression or induction of ectopic Wg expression).

We thank the reviewer for raising this important point. As outlined below, we quantified the fly experiments and conducted statistical analysis across multiple experimental datasets to further substantiate our claims. They are included in the updated manuscript.

Also, while the constructs used in the study were inserted into the same genomic landing sites to achieve comparable levels of expression of the various proteins, these reviewers would like to see data on the levels of expression of NEUR1 and 2 as well as the hybrid Notch ligands.

****Major comments****

Comment on fly wing disc experiments:

The authors study both the capability of two different mammalian E3 ubiquitin ligases, Neuralized-like 1 and 2 (mouse Neur1 and human NEUR2) to activate four different Notch receptors (DLL1 and 2, JAG1 and 2) in flies and mammalian cell culture system. In flies, they first analyse the capability of Drosophila Neur (as a positive control) and Neur1 and NEUR2 to

activate the various Notch ligands (based on wingless activation as a readout) in wild-type wings (where, Mind bomb 1, or Mib1 is the only E3 ligase for Notch ligands present) and Mib1 mutant wing discs (which lack any E3 ligands of Notch receptors). The authors then test four humanised, hybrid Notch ligands (all five N ligands bar Dll3 since the latter does not transactivate the Notch receptor) - where mammalian Notch ligands' intracellular domains, or ICDs, have been attached to fly DI (DI-Dll1, DI-Dll4, DI-JAG1, DI-JAG2) - for their capacity to mediate Mib1-dependent activation of Notch (with ectopic Wg expression in wing discs as its readout). They found that all 4 ligands can activate Notch in wild-type wings (where Mib1 is present), with DI-JAG2 being less effective than the other 3 hybrid ligands, implying that such hybrid, humanised ligands can be used in studying Notch pathway activation in Drosophila (thereby constituting a mixed/heterologous experimental system). The reviewers would like to get a comment as to the reason for the weaker activity of DI-JAG2 in this set-up?.

We do not have a definitive answer as to why the ICDs differ in their activity within Mib1-dependent signalling, since this question was not addressed in the scope of this work. However, our findings demonstrate that the hybrid ligands are functional in Drosophila and that their differential behavior in Neur-mediated signaling is not attributed to a trivial explanation, e. g. that the hybrid ligands generally display no activity. There are several potential explanations for these differences. One possibility is variations in position, arrangement, or number of targeted lysines among the ICDs. These lysines serve as substrates for ubiquitylation and determine the rate of endocytosis, which in turn impacts the signaling activity of the corresponding ligand/hybrid. Another plausible explanation is differences in affinity of the binding sites of Mib1, which would similarly result in variations in ubiquitylation and endocytosis rates. Regardless, we emphasize that resolving this question does not affect any of the conclusions of the manuscript and is a topic worth investigating in the future.

Also, the reviewers would like to get a comment as to why was not a Neur mutant set-up used, only Mib1 mutant discs?

Neur is only expressed at a very late stage in wing development and is restricted to specific single cells (sensory organ precursors). Consequently, even if mutants were present, their impact would be limited to these cells. We have added a comment in the revised manuscript addressing these considerations.

The authors then found that only two of these hybrid ligands - DI-DLL1 and DI-JAG1 but not DI-DLL4 or DI-JAG2 - can be used to activate Notch in the above wing assay when Mib1 was mutant. This is consistent with the fact that the NxxN-based Neuralized Binding motif (NBM) is present in DLL1 and JAG1 only. Using the wing paradigm, the authors also show by either mutating the full NBM (NxxN) in DLL1 or changing the cryptic, "weak" NBM in DLL4 (containing NxxD sequence) into "full/strong" NxxN one that the NBM in the various Notch ligands is required and sufficient for activation of the Notch pathway.

Overall, the fly experiments are convincing in showing differential activation of Notch ligands. However, no statistical analysis of the experimental variation in these studies - neither for the number of wing discs analysed per (hybrid) Notch ligand tested nor the extent of a given experimental manipulation's effect is included. We deem that if the images presented in Figures 2 and 3 are truly representative, this needs to be made explicitly clear.

We thank the reviewer for their positive evaluation of our work and for the constructive comments, which we included into the updated manuscript.

To address this, we quantified the experiments conducted in the wing imaginal discs of *Drosophila*. In the revised manuscript we have added quantification of the experiments conducted in the wing imaginal discs of *Drosophila* by measuring the over-proliferation of the wing disc along the dorsal-ventral axis relative to the anterior-posterior axis and the length of the Wg expression lines. Statistical analysis was performed to demonstrate statistical significance across n=5 experiments for each sample. They are now included in the updated manuscript in Fig. 2. G,H, Appendix Fig. 2SC, Fig. 3D, Appendix Fig. 3SA, Fig. 5D.

Comment on fly embryonic Delta neurogenic phenotype's rescue experiments by replacing DI with the hybrid ligands:

The authors analysed the capacity of the ICDs of the mammalian ligands to rescue the DI phenotype in *Drosophila*, ie. their activation capability at the organismal level. This was achieved by generating knock-in alleles expressing the hybrid ligands in place of DI. The notion that only NBM-containing hybrid ligands was strengthened by this analysis since it showed that only NBM-containing hybrid ligands - DI-DLL1 and DI-JAG1 - but not DI-DLL4 nor DI-JAG2 rescued the DI neurogenic embryonic lethal phenotype. Since this experimental set-up relied on the endogenous *Drosophila* E3 ligases for activating the Notch ligands, the capacity of mammalian NEUR1 and 2 proteins to complement the capacity of the hybrid ligands to activate Notch to activate these ligands was not addressed. Please comment as to the reasons for this apparent omission and if such an analysis lies beyond the scope of current work, what would be the expected results of such experiment in the light of other experiments conducted in the course of this work?

Testing whether mammalian *Neur1* and *NEURL1B* can replace *Drosophila* *Neur* in an endogenous setting is an intriguing question; however, it falls outside the focus of this study. Performing such an experiment would be highly challenging due to complex architecture of *Neur* gene in *Drosophila*. Additionally, we believe it is highly unlikely that the mammalian *NEURLs* proteins would fully compensate for the loss of function in a *Drosophila* *Neur* mutant.

Journal-agnostic peer review: evaluate the paper as it stands independently from potential journal fit.

Are the claims and the conclusions supported by the data or do they require additional experiments or analyses to support them?

Generally yes, but please see the above comments on the absence of statistical analysis of reproducibility/ variation (if any) in fly wing disc experiments.

****Reviewer's additional recommendations:****

To publish in a higher-ranking journal, the co-localisation analyses of Notch ligands and its various E3 ubiquitin ligases studied probably needs to be replaced by a more rigorous, ideally FRET-based approach.

We thank the reviewer for the comment. The co-localization assay is quite a robust and functional approach, as it provides clear evidence that endocytosis into a different compartment has occurred with functional ligands, as opposed to non-functional ligands. This serves as a quantitative and rigorous indicator for functional differences between these ligand types.

Following the recommendation for a more direct interaction readout, we performed additional co-immunoprecipitation (co-IP) assays in our cell culture system to complement the existing data and added the data to the manuscript (New section, and Appendix Fig. 6S). Surprisingly, all tested ligand variants co-immunoprecipitated with NEURL1, and neither removal of the NxxN motif from DLL1 and JAG1 nor its introducing into DLL4 significantly affected the binding interaction. These results suggest that, unlike the situation in *Drosophila* where mutation of the NBM in DI completely abolishes binding to Neur (ref 9), mammalian NEURL1 binds Notch ligands through a more complex mechanism. Thus, although the NxxN motif is essential for ligand activation and proper localization in mammalian cells, additional interaction sites or adaptor proteins are likely involved in stabilizing the ligand-NEURL1 complex.

Since previous studies have shown that the Notch ligands are (mostly) poly- or mono-ubiquitylated by the E3 ubiquitin ligases Mib and the NEUR proteins, ideally, this or its follow-up study would benefit from analysis of the ubiquitylation status of the various hybrid Notch ligands.

We thank the reviewer for the suggestion. The ubiquitylation pattern by Neur1 is beyond the topic of the current manuscript.

Also, it would be useful to show the strength of interaction between the hybrid Notch ligands and NEUR1 and NEUR2 by using a co-immunoprecipitation based approach.

As suggested by the reviewer, we performed co-IP as discussed in the previous response.

Please request additional experiments only if they are essential for the conclusions. Alternatively, ask the authors to qualify their claims as preliminary or speculative, or to remove them altogether.

These reviewers do not strictly request any further experiments. However, since the mammalian NEUR2 could not be studied in cell cultures of U2OS cells due to its toxicity, we would like the authors to explain the choice of this cell line. Perhaps a cell line whose viability is not impaired by NEUR2 should be (or should have been) used?

The decision not to use other cell lines was based on several strict experimental requirements. The most stringent requirement was the need to generate a MIB1 knockout cell line, as MIB1 strongly activates all ligands. The availability of having MIB1 KO U2OS cells enabled these experiments.

If you have constructive further reaching suggestions that could significantly improve the study but would open new lines of investigations, please label them as "OPTIONAL".

As mentioned above, the NEUR2's capacity to activate the hybrid ligands in U2OS cells could not be addressed due to its toxicity. A more optimal cell line will have to be used in follow-up studies.

Also, these findings ultimately warrant in vivo studies using mice to definitively ascertain whether they also hold equally true there.

Are the suggested experiments realistic in terms of time and resources? It would help if you could add an estimated time investment for substantial experiments.

The suggested experiments are optional apart from statistical analysis of variation (if any) in the fly wing disc experiments. If there is no (apparently significant) variation in these data, this needs to be explicitly stated.

We thank the reviewer for their thoughtful assessment. As pointed above, we conducted the requested quantification and statistical analysis to show significance of the results across multiple repeats.

Are the data and the methods presented in such a way that they can be reproduced?

Generally yes, but see above about the lack of statistical data on the variation (if any).

Are the experiments adequately replicated and statistical analysis adequate?

Generally yes, but again, please see above about the lack of statistical data on the variation (if any).

****Minor comments****

Comment#1 (on the abstract and introduction):

In the Abstract, the authors state that there are four Notch ligands in mammals (lines 21 and 22):

"Thus, it is unclear how NEURL proteins regulate the four mammalian Notch ligands".

In the Introduction, they correctly state that there are five Notch ligands in mammals (lines 38 and 39):

„In mammals, there are five ligands, three from the Delta-like (Dll) family (Dll1, Dll3, Dll4), and two from the Jagged (Jag) family (Jag1 and Jag2)."

There are five Notch ligands in mammals (Dll1, Dll3, Dll4, Jag1, Jag2), and it is obvious that the authors are very well aware of this (they state in lines 146-147):

"We excluded the ICD of DLL3 since it is not a ligand capable of trans-activation of Notch" (the four ligands included were Dll1, Dll4, Jag1 and Jag2)."

Therefore, a clarification is required in the part of Abstract (i.e lines 21 and 22) - did the authors mean the four mammalian Notch ligands they actually studied (i.e Dll1, Dll4, Jag1, Jag2) or is there an oversight and the authors actually intended to write "the five Notch ligands in mammals".? In either case, a correction is required in this reviewer's opinion.

We are fully aware of this point, and we updated the manuscript to address this by rephrasing the abstract to avoid misinterpretation.

Specific experimental issues that are easily addressable.

NEUR2 could not be studied in mammalian cell cultures due to its toxicity in the U2OS cell line, the one used by the authors. The use of another cell line would not be probably overly time-consuming; however, if this experiment lies outside the scope of current work, we would like to hear the authors' comment on this matter.

This is addressed in the response above.

Are prior studies referenced appropriately?

Generally yes, but four prior studies go unmentioned: the two 2001 mouse Neur1 knock-out studies reporting no Notch-like developmental phenotype (Ruan et al, PNAS; Vollrath et al, Mol Cell Biol), the 2002 study of mouse, rat and human NEUR1 expression, subcellular localisation (Timmusk et al, Mol Cell Neuroscience) and the 2009 cell culture-based study of NEUR2's interaction with DLL1 and DLL4 (Rullinkov et al, BBRC).

The non-requirement of NEUR1 and 2 proteins in mammalian developmental Notch signalling could partly be explained by the fact that NEUR1 is not highly expressed during mouse embryonic/foetal development - its expression becomes considerably more pronounced only postnatally (Timmusk et al, 2002).

We thank the reviewer for these suggestions. We incorporated almost all these references into the introduction and discussed the low expression of Neurl1 during development as a possible reason for the non-requirement in this context. The only reference we chose not to include was 2001 Vollrath et al, Mol Cell Biol study on knock-out mice. We found that the article had an addendum in proof acknowledging discrepancies in expression patterns and reproductive behaviors compared to another independently published study; as such, we decided not to include it in this manuscript at this time.

Are the text and figures clear and accurate?

Yes. These reviewers find the cartoon-based explanations of the experimental set-up in each figure helpful for enhancing the manuscript's overall clarity.

We thank the reviewers for the positive feedback.

Do you have suggestions that would help the authors improve the presentation of their data and conclusions?

Please see above about the lack of statistical data on the variation (if any) in fly wing disc experiments and referencing of the 4 papers that are currently excluded.

These were corrected in the revised version.

Reviewer #3 (Significance (Required)):

2. Significance

Provide contextual information to readers (editors and researchers) about the novelty of the study, its value for the field and the communities that might be interested.

The following aspects are important:

General assessment: provide a summary of the strengths and limitations of the study. What are the strongest and most important aspects? What aspects of the study should be improved or could be developed?

This study uses the amenability of *Drosophila* to study the mammalian NEUR proteins' (NEUR1 and NEUR2) activity upon Notch ligands using hybrid Notch ligands containing mammalian ICDs (intracellular domains) fused to the extracellular domain of *Drosophila* Delta (DI). It confirms and extends prior studies showing that Notch ligands can be (strongly) activated only by the E3 ubiquitin ligases containing the Neuralized Binding Motif (NBM).

We respectfully disagree with the reviewer's assessment on this point. Our study is the first to demonstrate that Neurl proteins differentially activate DLL1 and JAG1, but not DLL4 and JAG2.

However, since this study was based on using hybrid ligands containing mammalian ICDs of Notch ligands fused to the extracellular domain of *Drosophila* Delta (DI), it is somewhat artificial. While NEUR1 was also studied in mammalian cell cultures (but not NEUR2 due to its toxicity), only an in vivo study using mice expressing with systematic changes to the Notch ligands' NBM will definitively reveal whether the conclusions reached by the authors hold true in vivo in a non-heterologous system.

We firmly believe that our combined 'humanized fly' model and quantitative cell culture assay represents an innovative and rigorous approach for testing humanized proteins in in-vivo settings, without the need for extensive mouse genetics. The conclusions of our experiments should not be dismissed solely based on the grounds of "not being performed in mice," as this would undermine much of current scientific research.

Advance: compare the study to the closest related results in the literature or highlight results reported for the first time to your knowledge; does the study extend the knowledge in the field and in which way? Describe the nature of the advance and the resulting insights (for example: conceptual, technical, clinical, mechanistic, functional,...).

The study's advances are chiefly mechanistic and functional since they show more definitively that the reason underlying the differing activation of four mammalian Notch ligands by mammalian NEUR1 and NEUR2 is mostly based upon the presence or otherwise of a conserved Neuralized Binding Motif, NBM.

Audience: describe the type of audience ("specialised", "broad", "basic research", "translational/clinical", etc...) that will be interested or influenced by this research; how will this research be used by others; will it be of interest beyond the specific field?

The audience for this study is the research studying the Notch signalling pathway. Since dysregulation of this pathway is implicated in a number of devastating diseases, any improved understanding of its mechanistic underpinnings could in the long run lead to better therapeutic management of diseases with significant involvement of malfunctioning Notch signalling.

Please define your field of expertise with a few keywords to help the authors contextualize your point of view. Indicate if there are any parts of the paper that you do not have sufficient expertise to evaluate.

Molecular biology, molecular neuroscience, developmental biology, cell-cell signalling, Notch signalling. All parts of the manuscript fall within our expertise.

Dear Prof. Sprinzak,

Thank you for submitting your revised manuscript. It has now been seen by one of the original referees.

As you will see, referee finds that the study is significantly improved during revision and recommend publication. However, the editorial points below need to be addressed before I can accept the manuscript.

- We note the following regarding the manuscript format: the manuscript currently contains figures. Figures need to be removed from the text file as they are uploaded separately. The legends should go at the very end of the manuscript, but plain text should be provided, not text boxes.
- Please provide 3-5 keywords for your study. These will be visible in the html version of the paper and on PubMed and will help increase the discoverability of your work.
- As per our guidelines, please add a 'Data Availability Section', where datasets and computer code that were generated in the reported study should be listed in a structured manner and placed after the Methods section. If your study does not include datasets, please insert the following statement: This study includes no data deposited in external repositories (please see <https://www.embopress.org/page/journal/14693178/authorguide#dataavailability> for further information).
- Please add the title 'Disclosure and Competing Interests Statement' before the statement "The authors declare that they have no conflict of interest".
- As per our format requirements, in the reference list, citations should be listed in alphabetical order and then chronologically, with the authors' surnames and initials inverted; where there are more than 10 authors on a paper, 10 will be listed, followed by 'et al.'. Please see <https://www.embopress.org/page/journal/14693178/authorguide#referencesformat>
- The funding information needs to be a part of Acknowledgments (Funding section heading should be removed).
- We note that the following figure panels have not been called out individually in the text: 2GH, 3D, 7D-H.
- We note the following regarding the Appendix file: it is currently missing Appendix Fig. S1 and Appendix Fig. S4. The file needs to be submitted in PDF format and needs a title page with a Table of Contents with a list of items and their page numbers.
- Reagents & Tools table needs to be removed from the manuscript text and needs to be uploaded as a separate file.
- We note the following regarding source data:
 - o A single zipped file per figure needs to be submitted.
 - o We note that some files are in .dzi format, which we cannot open. As per our source data guidelines, images should be provided in a common format that preserve details (e.g. .tif files) (https://www.embopress.org/pb-assets/embosite/Source_Data_Guidelines.pdf).
- We note the following regarding the Data Availability section:
 - o A link that directly resolves to the dataset with the accession number phs000424.v10.p2 needs to be provided instead of the link to the homepage of the repository.
 - o The Data Availability section is reserved for the primary datasets generated in the study. Therefore, please move the reference to the existing (re-analyzed in this study) Allen Brain Cell Atlas datasets to the relevant parts of the manuscript text.
- We note that reference #37 (Kuintzle et al. 2023) has meanwhile been published (PMID: 39751380). Please update the reference list accordingly.
- The manuscript sections should be in the following order: Title page - Abstract & Keywords - Introduction - Results - Discussion - Methods - Data Availability - Acknowledgments - Disclosure Statement & Competing Interests - References - Figure Legends - (Main Tables with legends if applicable) - Expanded View Figure Legends.
- There are two unlabeled tables in the manuscript. They should either be named Table 1 and 2, or uploaded separately as Table EV1 and EV2.
- During our routine image analyses, we note a reuse between this manuscript (Figure 7A) and <https://doi.org/10.1186/s12915-023-01759-z> (Figure 1B). We discourage republishing previously published images. We recommend citing the study pointing out the specific figure panel instead. Please contact us if you would like to discuss this point further.
- Our production/data editors have asked you to clarify several points in the figure legends - Figure Legends (main + EV):
 - o Please note that the exact p values are not provided in the legends of figures 2G, H; 4B-D; 5D-G; 6B'-D'
 - o Please note that the error bars are not defined in the legends of figures 2G, H; 4B-D; 5D-G; 6B'-D'
- Papers published in EMBO Reports include a 'synopsis' and 'bullet points' to further enhance discoverability. Both are displayed on the html version of the paper and are freely accessible to all readers. The synopsis includes a short standfirst summarizing the study in 1 or 2 sentences (max 35 words) that summarize the paper and are provided by the authors and streamlined by the handling editor. I would therefore ask you to include your synopsis blurb and 3-5 bullet points listing the key experimental findings.
- In addition, please provide an image for the synopsis. This image should provide a rapid overview of the question addressed in the study but still needs to be kept fairly modest since the image size cannot exceed 550 (width) x 300-600 (height) pixels.

Thank you again for giving us to consider your manuscript for EMBO Reports, I look forward to your minor revision.

Kind regards,

Deniz Senyilmaz Tiebe

--

Deniz Senyilmaz Tiebe, PhD
Senior Scientific Editor
EMBO Reports

Referee #1:

The authors are providing a revised manuscript that is further improved and responsive to feedback, including through a significant amount of additional work.

Referee #1:

The authors are providing a revised manuscript that is further improved and responsive to feedback, including through a significant amount of additional work.

Rev_Com_number: RC-2024-02705

New_manu_number: EMBOR-2024-61042V2

Corr_author: Sprinzak

Title: Neuralized-like proteins differentially activate Notch ligands

Dear Editors,

Please find below point-by-point response to the editorial corrections requested.

- We note the following regarding the manuscript format: the manuscript currently contains figures. Figures need to be removed from the text file as they are uploaded separately. The legends should go at the very end of the manuscript, but plain text should be provided, not text boxes.

Manuscript reorganized as requested

- Please provide 3-5 keywords for your study. These will be visible in the html version of the paper and on PubMed and will help increase the discoverability of your work.

Keywords provided below the abstract

Notch Signaling

Neuralized

E3 ubiquitin ligases

DSL ligands

- As per our guidelines, please add a 'Data Availability Section', where datasets and computer code that were generated in the reported study should be listed in a structured manner and placed after the Methods section. If your study does not include datasets, please insert the following statement: This study includes no data deposited in external repositories (please see <https://www.embopress.org/page/journal/14693178/authorguide#dataavailability> for further information).

Data availability section is provided.

- Please add the title 'Disclosure and Competing Interests Statement' before the statement "The authors declare that they have no conflict of interest".

Provided.

- As per our format requirements, in the reference list, citations should be listed in alphabetical order and then chronologically, with the authors' surnames and initials inverted; where there are more than 10 authors on a paper, 10 will be listed, followed by 'et al.'. Please see <https://www.embopress.org/page/journal/14693178/authorguide#referencesformat>

Citation format changed according to requirement.

- The funding information needs to be a part of Acknowledgments (Funding section heading should be removed).

Changed as requested.

- We note that the following figure panels have not been called out individually in the text: 2GH, 3D, 7D-H.

All panels are now called in the main text.

- We note the following regarding the Appendix file: it is currently missing Appendix Fig. S1 and Appendix Fig. S4. The file needs to be submitted in PDF format and needs a title page with a Table of Contents with a list of items and their page numbers.

Appendix Figure names changed as requested.

- Reagents & Tools table needs to be removed from the manuscript text and needs to be uploaded as a separate file.

Provided separately.

- We note the following regarding source data:
 - o A single zipped file per figure needs to be submitted.

Now submitted as single files.

- o We note that some files are in .czi format, which we cannot open. As per our source data guidelines, images should be provided in a common format that preserve details (e.g. .tif files) (https://www.embopress.org/pb-assets/embopress-site/Source_Data_Guidelines.pdf).

Submitted as Tiff files.

- We note the following regarding the Data Availability section:

o A link that directly resolves to the dataset with the accession number phs000424.v10.p2 needs to be provided instead of the link to the homepage of the repository.

Changed as requested

o The Data Availability section is reserved for the primary datasets generated in the study. Therefore, please move the reference to the existing (re-analyzed in this study) Allen Brain Cell Atlas datasets to the relevant parts of the manuscript text.

Changed as requested.

- We note that reference #37 (Kuintzle et al. 2023) has meanwhile been published (PMID: 39751380). Please update the reference list accordingly.

Changed to the published reference.

- The manuscript sections should be in the following order: Title page - Abstract & Keywords - Introduction - Results - Discussion - Methods - Data Availability - Acknowledgments - Disclosure Statement & Competing Interests - References - Figure Legends - (Main Tables with legends if applicable) - Expanded View Figure Legends.

Changed to the requested format.

- There are two unlabeled tables in the manuscript. They should either be named Table 1 and 2, or uploaded separately as Table EV1 and EV2.

Tables named as Table 1 and Table 2.

- During our routine image analyses, we note a reuse between this manuscript (Figure 7A) and <https://doi.org/10.1186/s12915-023-01759-z> (Figure 1B). We discourage republishing previously published images. We recommend citing the study pointing out the specific figure panel instead. Please contact us if you would like to discuss this point further.

Panel 7A (a control image) was replaced by an unpublished image.

- Our production/data editors have asked you to clarify several points in the figure legends - Figure Legends (main + EV):

- o Please note that the exact p values are not provided in the legends of figures 2G, H; 4B-D; 5D-G; 6B'-D'
- o Please note that the error bars are not defined in the legends of figures 2G, H; 4B-D; 5D-G; 6B'-D'

Actual p-values are now provided within the figures. Error bars are defined as requested.

- Papers published in EMBO Reports include a 'synopsis' and 'bullet points' to further enhance discoverability. Both are displayed on the html version of the paper and are freely accessible to all readers. The synopsis includes a short standfirst summarizing the study in 1 or 2 sentences (max 35 words) that summarize the paper and are provided by the authors and streamlined by the handling editor. I would therefore ask you to include your synopsis blurb and 3-5 bullet points listing the key experimental findings.

We now provide a synopsis blurb and bullet points:

The NEURL E3 ubiquitin ligases selectively activate the Notch ligands DLL1 and JAG1, but not DLL4 and JAG2. This differential regulation reveals a motif-dependent mechanism that may potentially explain NEURL's limited developmental role in mammals.

- **NEURL proteins** differentially activate Notch ligands DLL1 and JAG1, but not DLL4 and JAG2.
- NEURL dependent activation requires an NxxN motif in the Notch ligand intracellular domain.
- Differential activity by NEURL proteins can potentially explain their limited developmental activity in mammals compared to flies.

- In addition, please provide an image for the synopsis. This image should provide a rapid overview of the question addressed in the study but still needs to be kept fairly modest since the image size cannot exceed 550 (width) x 300-600 (height) pixels.

We now provide an image for the synopsis

Prof. David Sprinzak
University of Tel Aviv
Biochemistry
Sherman building, room 508
The George S. Wise faculty of life sciences
Tel Aviv
Israel

Dear Prof. Sprinzak,

I am very pleased to accept your manuscript for publication in the next available issue of EMBO reports. Thank you for your contribution to our journal.

Yours sincerely,
